# PointCHR: Point Cloud Analysis via Curvature-Aware Hyperbolic Rectification

Xinxing Yu [1]  Liying Yang [1]  Hao Mo [1]  Hui Ma [* 2 3]  Fang Kai [1]  Ajian Liu [1 4]  Yanyan Liang [* 1]

## Abstract

High-curvature regions in 3D point clouds encapsulate critical fine-grained geometric semantics yet exhibit a distinct long-tail sparsity in their spatial distribution. The inherent limitations of polynomial volume growth in Euclidean space frequently render these intricate geometric features challenging to adequately resolve within a uniform-scale feature space. Consequently, these regions are frequently overshadowed by smooth global features dominated by low-curvature regions, thereby limiting the discriminative capacity of the network. To address this issue, we propose PointCHR, a curvature-aware hyperbolic rectification (CHR) for point cloud analysis. Utilising the property of exponential volume expansion in the vicinity of hyperbolic manifolds, CHR presents a learnable curvature-guided radial rectification mechanism. By adaptively projecting high-curvature points towards boundary regions endowed with larger effective embedding capacities, PointCHR effectively mitigates the representation crowding problem inherent in Euclidean settings. Extensive experimentation has demonstrated that PointCHR significantly enhances the ability of backbone to capture fine-grained geometric details, achieving state-of-the-art performance across multiple benchmarks.

## 1. Introduction

3D understanding serves as the cornerstone for numerous applications, such as detection(Liu et al., 2025; Schreier

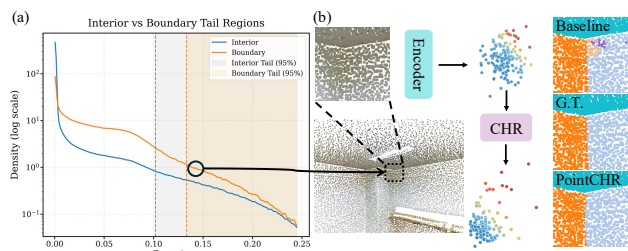

*Figure 1.* Illustration of the curvature imbalance challenge. (a) Quantitative analysis shows the long-tailed nature of curvature distribution in S3DIS datast. (b) In complex corner regions, conventional encoders fail to handle such tail samples, leading to boundary artifacts and noise. Our CHR addresses this by incorporating geometric priors for feature optimization, successfully restoring crisp boundaries consistent with the Ground Truth.

et al., 2023),reconstruction (Zhu et al., 2023a;b; Yang et al., 2023; Yu et al., 2025; Wang et al., 2026) and generation (Yang et al., 2025a; 2026; Han et al., 2024a; 2025a). Recent advances in point cloud deep learning have led to a series of effective architectures (Qian et al., 2022; Wang et al., 2024), which show capability in learning global representations. Notwithstanding these advances, most methods aggregate local neighborhoods within Euclidean feature spaces and implicitly presume a homogeneous and universally adequate representational capacity across all geometric primitives.

However, the prevailing Euclidean assumption overlooks the intrinsic heterogeneity of three dimension (3D) geometries. As shown in Figure 1(a), local curvature exhibits a long-tail distribution: most points lie on low-curvature surfaces , whereas high-curvature structures, such as corners, edges, and fine-grained details are sparsely distributed. Although rare, these regions often carry informative fine-grained geometric cues vital for accurate recognition.

This geometric heterogeneity imposes a structural impediment on Euclidean embeddings employing isotropic scaling. Constrained by the polynomial volume growth of Euclidean space, sparse yet topologically complex high-curvature structures are compelled to vie for a finite representational budget against ubiquitous low-curvature surfaces. This capacity imbalance precipitates representation crowding, wherein fine-grained geometric cues are com-

---

[*]Corresponding author [1]School of Computer Science and Engineering, Macau University of Science and Technology, Macau, China [2]School of Computing and Information Technology, Great Bay University, Dongguan, China [3]Shenzhen International Graduate School, Tsinghua University, Shenzhen, China [4]MAIS, The Institute of Automation of the Chinese Academy of Sciences, Beijing, China. Correspondence to: Hui Ma <mahuilight@gmail.com>, Yanyan Liang <yyliang@must.edu.mo>.

*Proceedings of the 43rd International Conference on Machine Learning*, Seoul, South Korea. PMLR 306, 2026. Copyright 2026 by the author(s).

pressed into restricted embedding subspaces and subsumed by dominant global patterns. As evidenced in Figure 1(b), such crowding induces a feature collapse of high-curvature regions into indistinguishable latent neighborhoods, thereby eroding class separability and compromising boundary localization. These observations indicate that the fundamental limitation of existing methods stems from the inefficient allocation of representational resources by isotropic embedding strategies across regions of varying geometric density. This necessitates a geometry-aware reparameterization mechanism to adaptively redistribute representational capacity commensurate with local geometric complexity, thereby fundamentally circumventing the crowding bottleneck.

To surmount this bottleneck, we propose aligning the feature space with the intrinsic geometric complexity of the data. Hyperbolic geometry, characterized by constant negative curvature, provides an optimal inductive bias for non-uniform structures. Diverging from Euclidean space and its polynomial growth limitations, hyperbolic space facilitates exponential volume expansion. This property generates immense representational capacity near the boundary and enables the model to unfold crowded high-curvature details into distinct high-resolution regions while preserving global structural integrity. By projecting complex geometric cues toward the boundary while retaining simpler features near the center, hyperbolic embeddings naturally accommodate the long-tailed geometric complexity of 3D point clouds and provide a principled solution to the crowding problem.

However, this direct adaptation is hindered by three structural impediments. First, structural heterogeneity: as shown in Figure 1(a), the semantic ambiguity arising from the entanglement of object boundaries and fine textures within high-curvature regions renders static curvature priors ineffective. Second, feature misalignment: the discrepancy between Euclidean features and the intrinsic geometric metric of the manifold restricts the effective activation of hyperbolic hierarchical capacity. Third, optimization instability: the asymptotic nature of boundary regions predisposes tangent-space transformations to gradient vanishing or numerical overflow, obstructing stable model convergence.

To address these challenges, we propose **PointCHR**, a curvature aware hyperbolic rectification for point cloud analysis comprising three integral components, namely Hyperbolic Semantic Transformation (**HST**), Point-wise Curvature-Adaptive Perception (**PCP**), and Closed-Form Geodesic Dilation (**CGD**). Specifically, targeting structural heterogeneity, PCP discards static heuristics in favor of leveraging curvature as geometric guidance fused with local context to dynamically learn point-wise radial scaling factors, which achieves a data-driven and discriminative disentanglement of object boundaries and fine textures. Subsequently, to resolve feature misalignment, HST projects Euclidean back-

bone features onto the Poincaré ball and employs Möbius linear transformations to reconstruct the semantic space on the manifold, thereby explicitly aligning the feature distribution with intrinsic hyperbolic geometry. Finally, to surmount optimization instability, CGD directly modulates geodesic distances based on the learned scaling factors via a tangent-space-free closed-form solution. This mechanism effectively circumvents the risk of numerical collapse in asymptotic boundary regions and enables the model to robustly unfold crowded high-curvature features into the high-capacity regions of the hyperbolic space.

In summary, our contributions are as follows:

- To the best of our knowledge, PointCHR constitutes the pioneering attempt to synergize intrinsic curvature cues with hyperbolic feature learning for point cloud analysis. This effectively bridges the gap between geometric topology and representation learning to offer a principled solution for representation crowding.

- We design a unified rectification pipeline integrating HST, PCP, and CGD to systematically resolve structural impediments. These components collaboratively correct feature misalignment, disentangle structural heterogeneity via adaptive scaling, and ensure optimization stability near the asymptotic boundary.

- Extensive experiments demonstrate that our method achieves state-of-the-art (SOTA) performance on standard benchmarks. Furthermore, qualitative results verify that PointCHR significantly enhances boundary delineation in geometrically complex regions.

## 2. Related Work

**3D Point Cloud Analysis.** Modern approaches generally follow a sample-and-group paradigm to aggregate local contexts. Mainstream backbones encode neighborhoods via convolution-based operators (Zhang et al., 2022; 2024; 2025c; Lin et al., 2023; Deng et al., 2023; 2024; Zheng et al., 2024; Li et al., 2024; Zha et al., 2025a;b; Zeng et al., 2025), graph attention (Wang et al., 2019; Xu et al., 2021b;a; Yin et al., 2024; Koch et al., 2024), Transformer (Vaswani et al., 2017) architectures (Zhao et al., 2021; Park et al., 2022; Wu et al., 2022; Park et al., 2023; Luo et al., 2024; Wu et al., 2024; 2025; Thomas et al., 2024; Xu et al., 2024; Kolodiazhnyi et al., 2024; Yang et al., 2025b; Han et al., 2025b), or recent Mamba-based models (Gu & Dao, 2024; Liang et al., 2024; Han et al., 2024b; Zhang et al., 2025b; Li et al., 2025; Bahri et al., 2025; LinshuangDiao et al., 2025; Yu et al., 2026b;a). To further enhance geometric sensitivity, several works incorporate explicit priors. RepSurf (Ran et al., 2022), CurveNet (Xiang et al., 2021) and Canon-Net (Friedmann & Werman, 2025) augment inputs with

attributes like curvature, while RS-CNN (Liu et al., 2019) and Point-NN (Zhang et al., 2023) exploit topological relations. Others employ geometry-aware attention (Qin et al., 2022; Xu et al., 2021c) or boundary-specific contrastive learning (Tang et al., 2022) to focus on complex regions. While these methods have achieved success in capturing shape features, they rely on isotropic Euclidean embeddings. However, constrained by the polynomial volume growth of Euclidean space, such approaches struggle to accommodate the expansive information of complex local geometries. This bottleneck induces a representation crowding effect, where fine-grained semantics in high-curvature regions are inevitably submerged by the dominant features of smooth surfaces, limiting the discriminative power of the network on critical boundary details.

**Hyperbolic Deep Learning.** Hyperbolic geometry has emerged as a powerful paradigm for modeling data with latent hierarchical structures and exponential volume growth. Seminal works (Nickel & Kiela, 2017; Ganea et al., 2018) introduced Poincaré embeddings and hyperbolic neural networks, demonstrating superior capacity over Euclidean counterparts in preserving semantic hierarchies with lower dimensionality. These advantages have been successfully extended to graph representation learning (Chami et al., 2019) and computer vision tasks, such as classification (Khrulkov et al., 2020; Montanaro et al., 2022; Feng et al., 2025) and zero-shot recognition (Liu et al., 2020). Recent advances have further integrated hyperbolic geometry into modern architectures. For instance, Hyperbolic Vision Transformers (Ermolov et al., 2022) combine Euclidean and hyperbolic features to capture diverse visual semantics, while optimization strategies like clipped hyperbolic classifiers (Guo et al., 2022) have been proposed to enhance training stability. However, the application of hyperbolic geometry to 3D point cloud analysis remains underexplored. Specifically, leveraging the exponential capacity of hyperbolic boundary regions to alleviate the feature crowding of high-curvature geometric primitives remains a promising yet underexplored avenue in 3D analysis.

## 3. Preliminaries

### 3.1. Curvature in Point Clouds

Given a point cloud $\mathcal{P} = \{\mathbf{p}_i \in \mathbb{R}^3\}_{i=1}^N$, the local geometric complexity around each point can be characterized by the variation of its spatial neighborhood. We adopt the widely used Principal Component Analysis (Hotelling, 1933) approach to estimate point-wise curvature (Pauly et al., 2002).

For each $\mathbf{p}_i$, we form the centered neighborhood matrix $\mathbf{X}_i \in \mathbb{R}^{k \times 3}$ using its $k$-nearest neighbors $\mathcal{N}(i)$:

$$\mathbf{X}_i = [(\mathbf{p}_{j_1} - \mathbf{p}_i), \dots, (\mathbf{p}_{j_k} - \mathbf{p}_i)]^\top. \quad (1)$$

The local geometric structure is encoded in the covariance matrix $\mathbf{C}_i = \frac{1}{k}\mathbf{X}_i^\top \mathbf{X}_i \in \mathbb{R}^{3 \times 3}$. Let $\lambda_{i,1} \geq \lambda_{i,2} \geq \lambda_{i,3} \geq 0$ denote the eigenvalues of $\mathbf{C}_i$. The smallest eigenvalue, $\lambda_{i,3}$, quantifies the variance along the surface normal, thereby reflecting the deviation of the local neighborhood from a planar surface. Following established conventions, we calculate the curvature estimate $\kappa_i$ at point $\mathbf{p}_i$ as:

$$\kappa_i = \frac{\lambda_{i,3}}{\sum_{m=1}^3 \lambda_{i,m} + \varepsilon}, \quad (2)$$

where $\varepsilon$ is a small constant for numerical stability.

### 3.2. Hyperbolic Geometry Basics

We employ the Poincaré ball model to represent hyperbolic geometry, as it is well-suited for gradient-based optimization. The Poincaré ball $(\mathbb{B}_c^d, g^{\mathbb{B}})$ of dimension $d$ with manifold curvature $-c$ ($c > 0$) is defined as the manifold $\mathbb{B}_c^d = \{\mathbf{x} \in \mathbb{R}^d : \|\mathbf{x}\|^2 < 1/c\}$, equipped with the Riemannian metric $g_\mathbf{x}^{\mathbb{B}} = \lambda_\mathbf{x}^2 g^E$, where $\lambda_\mathbf{x} = \frac{2}{1 - c\|\mathbf{x}\|^2}$ is the conformal factor and $g^E$ denotes the Euclidean metric.

While the theoretical domain is the open ball $\mathbb{B}_c^d$, numerical implementations require bounding the inputs away from the boundary to prevent arithmetic overflow, since $\operatorname{arctanh}(z) \to \infty$ as $z \to 1$. We therefore operate within a closed subset $\mathbb{B}_{c,\varepsilon}^d \subset \mathbb{B}_c^d$:

$$\mathbb{B}_{c,\varepsilon}^d = \left\{ \mathbf{x} \in \mathbb{R}^d \mid \|\mathbf{x}\| \leq \frac{1-\varepsilon}{\sqrt{c}} \right\}, \quad (3)$$

where $\varepsilon$ is a small constant margin.

An input feature vector $\mathbf{v} \in \mathbb{R}^d$ is mapped to the Poincaré ball via the exponential map $\operatorname{Exp}_\mathbf{0}^c$, and mapped back via the logarithmic map $\operatorname{Log}_{\mathbf{0},\varepsilon}^c$. The numerically stable formulations are given by:

$$\operatorname{Exp}_\mathbf{0}^c(\mathbf{v}) = \tanh(\sqrt{c}\|\mathbf{v}\|)\frac{\mathbf{v}}{\sqrt{c}\|\mathbf{v}\|}, \quad (4)$$

$$\operatorname{Log}_{\mathbf{0},\varepsilon}^c(\mathbf{x}) = \operatorname{arctanh}(\sqrt{c}\|\tilde{\mathbf{x}}\|)\frac{\tilde{\mathbf{x}}}{\sqrt{c}\|\tilde{\mathbf{x}}\|} \quad (5)$$

where $\tilde{\mathbf{x}} = \min\left(1, \frac{1-\varepsilon}{\sqrt{c}\|\mathbf{x}\|}\right)\mathbf{x}$. Note that for $\mathbf{v} \to \mathbf{0}$, the term $\frac{\tanh(v)}{v} \to 1$, ensuring the exponential map approximates the identity mapping near the origin. This mechanism allows us to lift standard Euclidean features onto the hyperbolic manifold to leverage its geometric properties.

## 4. Method

### 4.1. Overall Architecture

We formulate our framework as a mapping $\Phi : \{\mathcal{P}, \mathcal{F}\} \to \mathbf{Y}$, transforming the input point cloud $\mathcal{P}$ and auxiliary features $\mathcal{F} \in \mathbb{R}^{N \times M}$ into task-specific predictions $\mathbf{Y}$. As shown in Figure 2(a), the pipeline consists of three stages:

**Feature Initialization.** We first extract the structural prior $\boldsymbol{\kappa} \in \mathbb{R}^N$ and the initial semantic embedding $\mathbf{H}_{\text{in}} \in \mathbb{R}^{N \times C}$ via a foundational feature encoder $\mathcal{E}$:

$$\boldsymbol{\kappa} = \Psi_{\text{curv}}(\mathcal{P}), \quad \mathbf{H}_{\text{in}} = \mathcal{E}([\mathcal{P} \oplus \mathcal{F}]), \quad (6)$$

where $\Psi_{\text{curv}}$ denotes the point-wise curvature estimation and $\oplus$ indicates channel-wise concatenation.

**Curvature-Aware Hyperbolic Rectification (CHR).** The deep feature extraction is governed by the **CHR** $\mathcal{B}$. Formally, the backbone output $\mathbf{H}_{\text{out}}$ is derived as:

$$\mathbf{H}_{\text{out}} = \mathcal{B}(\mathbf{H}_{\text{in}} \mid \boldsymbol{\kappa}) \in \mathbb{R}^{N \times C'}. \quad (7)$$

The notation $(\cdot \mid \boldsymbol{\kappa})$ emphasizes that the deep feature transformation is strictly conditioned on the geometric prior $\boldsymbol{\kappa}$.

**Task Prediction.** Finally, the geometrically rectified representations $\mathbf{H}_{\text{out}}$ are projected onto the target label space via a task-specific decoder $\mathcal{T}$. Whether for classification or segmentation, the prediction $\mathbf{Y}$ is generated as:

$$\mathbf{Y} = \mathcal{T}(\mathbf{H}_{\text{out}}). \quad (8)$$

### 4.2. Curvature-Aware Hyperbolic Rectification (CHR)

Standard Euclidean networks often struggle to distinguish features in regions with high geometric complexity due to the lack of sufficient margin. To address this, we propose the CHR, which dynamically adjusts the hyperbolic embeddings based on local point geometry cues. The block consists of three stages: Hyperbolic Semantic Transformation, Point-wise Curvature-Adaptive Perception, and Closed-Form Geodesic Rescaling. The overall procedure is summarized in Algorithm 1.

**Hyperbolic Semantic Transformation (HST).** Let $\mathbf{h}_i^{in} \in \mathbb{R}^C$ denote the input feature vector for the $i$-th point in the ambient Euclidean space. To leverage the hyperbolic geometry, we first lift the input onto the Poincaré ball. By identifying the Euclidean input space with the tangent space at the origin $(T_{\mathbf{0}} \mathbb{B}_c^C \cong \mathbb{R}^C)$, we apply the exponential map:

$$\mathbf{z}_i^{(0)} = \text{Exp}_{\mathbf{0}}^c(\mathbf{h}_i^{in}) \in \mathbb{B}_c^C. \quad (9)$$

This operation projects the standard features onto the manifold, initializing the hyperbolic embedding process.

To facilitate expressive semantic feature extraction while strictly preserving the underlying hyperbolic geometry, we employ the Möbius linear layer. Standard Euclidean linear transformations are fundamentally incompatible with this objective, as they disregard the underlying conformal geometry of the Poincaré ball, thereby inevitably introducing severe geometric distortions. Consequently, we adopt the

---

**Algorithm 1** Curvature-Aware Hyperbolic Rectification

1: **Input:** Euclidean feature matrix $\mathbf{H}_{\text{in}} \in \mathbb{R}^{N \times C}$, Normalized point curvature $\boldsymbol{\kappa} \in \mathbb{R}^N$
2: **Hyperparameters:** Manifold curvature $c$, Dilation scale $\alpha$, Sensitivity $\gamma$, Stability margin $\varepsilon$
3: **Output:** Rectified Euclidean features $\mathbf{H}_{\text{out}} \in \mathbb{R}^{N \times C}$
4: *// 1. Hyperbolic Semantic Transformation*
5: $\mathbf{Z}^{(0)} \leftarrow \text{Exp}_{\mathbf{0}}^c(\mathbf{H}_{\text{in}})$
6: $\mathbf{Z}^{(1)} \leftarrow \text{MöbiusActivation}(\text{MöbiusLinear}(\mathbf{Z}^{(0)}))$
7: *// 2. Point-wise Curvature-Adaptive Perception*
8: $\mathbf{G} \leftarrow \text{Softplus}(\text{MLP}(\mathbf{H}_{\text{in}} \oplus \boldsymbol{\kappa}))$
9: $\mathbf{s} \leftarrow 1 + \alpha \cdot \mathbf{G} \odot (\boldsymbol{\kappa})^\gamma$
10: *// 3. Closed-Form Geodesic Dilation*
11: Initialize $\mathbf{Z}' \in \mathbb{B}_c^{N \times C}$
12: **for** each point $i \in \{1, \dots, N\}$ **do**
13: $\quad \mathbf{z}_i \leftarrow \mathbf{Z}_i^{(1)}$
14: $\quad \tilde{r}_i \leftarrow \max(\|\mathbf{z}_i\|_2, \varepsilon)$
15: $\quad d_i' \leftarrow s_i \cdot \text{arctanh}(\sqrt{c} \cdot \tilde{r}_i)$
16: $\quad \lambda_i \leftarrow \frac{1}{\sqrt{c}\tilde{r}_i} \tanh(d_i')$
17: $\quad \mathbf{Z}_i' \leftarrow \lambda_i \cdot \mathbf{z}_i$
18: **end for**
19: *// 4. Projection back to Euclidean Space*
20: $\mathbf{H}_{\text{out}} \leftarrow \text{LayerNorm}(\text{Log}_{\mathbf{0},\varepsilon}^c(\mathbf{Z}'))$
21: **return** $\mathbf{H}_{\text{out}}$

---

Möbius generalization of the linear map, which is defined via Möbius matrix multiplication ($\otimes_c$) and addition ($\oplus_c$):

$$\mathbf{z}_i' = \text{MöbiusLinear}(\mathbf{z}_i^{(0)}) = (W \otimes_c \mathbf{z}_i^{(0)}) \oplus_c \mathbf{b}, \quad (10)$$

where $W \in \mathbb{R}^{C \times C}$ is the learnable weight matrix and $\mathbf{b} \in \mathbb{B}_c^C$ is the hyperbolic bias.

To introduce non-linearity, we apply the Möbius variant of the GELU activation (Hendrycks, 2016), denoted as $\sigma^{\otimes_c}$. This operation ensure the resulting features remain well-defined on the manifold:

$$\mathbf{z}_i^{(1)} = \text{MöbiusActivation}(\mathbf{z}_i') = \sigma^{\otimes_c}(\mathbf{z}_i'). \quad (11)$$

By stacking these operations, the block effectively learns complex semantic interactions without degrading the hierarchical structure encoded in the hyperbolic embeddings.

**Point-wise Curvature-Adaptive Perception (PCP).** While the preceding semantic transformation effectively captures feature interactions, it operates in an isotropic manner, treating the hyperbolic space as a homogeneous medium. This overlooks a fundamental property of the Poincaré ball: exponential volume expansion. Specifically, the spatial capacity of the manifold increases exponentially as the distance from the origin grows. Consequently, regions near the boundary offer significantly larger embedding volume, making them ideal for resolving fine-grained details, whereas the origin is constrained and better suited for low-entropy information.

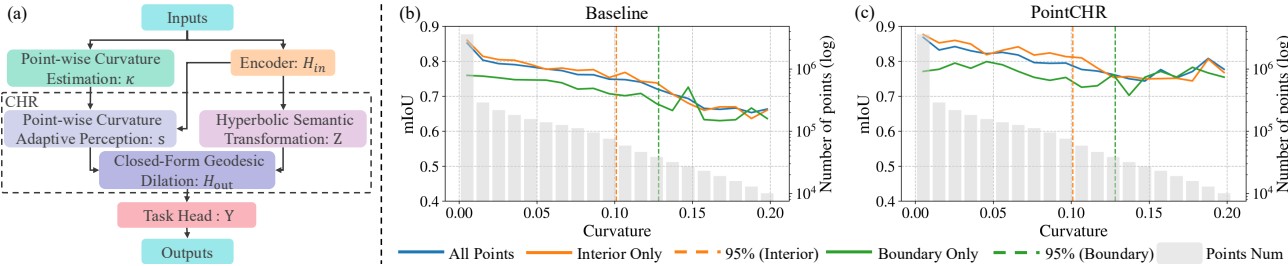

*Figure 2.* **(a)** Overview of the proposed PointCHR framework. **(b)** The Euclidean baseline struggles with this data imbalance, showing a sharp performance degradation in the high-curvature tail. **(c)** Our PointCHR effectively rectifies this issue. By leveraging hyperbolic geometry to handle structural complexity, our method prevents performance collapse in the tail, demonstrating superior robustness on rare hard samples. The background histograms visualize the long-tail distribution of point curvature in the S3DIS dataset, highlighting the sparsity of high-curvature samples.

We posit that the local geometry of 3D point clouds exhibits a latent hierarchy analogous to this spatial structure. Points situated on flat surfaces typically represent generic shape structures with low information density. Conversely, points in high-curvature regions encapsulate specific, high-frequency geometric details. To maximize representation efficiency, we propose to explicitly align this geometric complexity with the radial capacity of the manifold. Our objective is to distinguishably embed these features by pushing high-curvature points towards the spacious boundary while retaining flat points near the stable origin. Therefore, we introduce a mechanism to dynamically modulate the embedding of radial position based on point curvature cues.

Let $\kappa_i \in [0, 1]$ be the normalized local geometric curvature of the $i$-th point. Relying solely on raw point-wise curvature is insufficient, as the importance of a geometric feature often depends on its semantic context. Therefore, we propose a learnable gating function $\mathcal{G}_\phi$ to adaptively fuse the geometric cue with the Euclidean semantic feature $\mathbf{h}_i^{in}$. The point-wise curvature-aware gating factor $g_i$ is computed as:

$$g_i = \text{Softplus}\left(\mathcal{G}_\phi([\mathbf{h}_i^{in} \oplus \kappa_i])\right), \quad (12)$$

where $\oplus$ denotes channel-wise concatenation, and $\mathcal{G}_\phi$ is a lightweight learnable Multi-Layer Perceptron.

We then formulate the hierarchy scaling factor $s_i$, which determines the degree of radial dilation. This factor couples the learned semantic weight $g_i$ with the geometric prior:

$$s_i = 1 + \alpha \cdot g_i \cdot (\kappa_i)^\gamma. \quad (13)$$

Here, $\alpha > 0$ is a hyperparameter controlling the maximum dilation amplitude, and $\gamma \geq 1$ modulates the sensitivity to point-wise curvature variations.

**Closed-Form Geodesic Dilation (CGD).** Upon computing the hierarchy scaling factor $s_i$, we adjust the radial position of the embedding to reflect the associated geometric

hierarchy. We perform this operation via Möbius scalar multiplication, which scales the geodesic distance from the origin while preserving the angular orientation.

Recall that the geodesic distance between the origin and a point $\mathbf{z} \in \mathbb{B}_c^d$ is given by $d_\mathbb{B}(\mathbf{0}, \mathbf{z}) = \frac{1}{\sqrt{c}}\text{arctanh}(\sqrt{c}\|\mathbf{z}\|)$. We seek a dilated embedding $\mathbf{z}_i^{(2)}$ that maintains the direction of $\mathbf{z}_i^{(1)}$ but scales the geodesic distance of the vector by the factor $s_i$:

$$d_\mathbb{B}(\mathbf{0}, \mathbf{z}_i^{(2)}) = s_i \cdot d_\mathbb{B}(\mathbf{0}, \mathbf{z}_i^{(1)}). \quad (14)$$

To implement this efficiently, we leverage the closed-form analytical solution of the Möbius scalar multiplication:

$$\mathbf{z}_i^{(2)} = \frac{1}{\sqrt{c}} \tanh\left(s_i \cdot \text{arctanh}(\sqrt{c}\tilde{r}_i)\right) \frac{\mathbf{z}_i^{(1)}}{\tilde{r}_i}, \quad (15)$$

where $\tilde{r}_i = \max(\|\mathbf{z}_i^{(1)}\|, \varepsilon)$ ensures numerical stability.

This formulation provides a direct geometric interpretation of hierarchy adjustment. Furthermore, it naturally satisfies the constraints of the manifold: since the image of the $\tanh(\cdot)$ function is strictly $(-1, 1)$, the norm of the rectified embedding $\|\mathbf{z}_i^{(2)}\|$ is intrinsically bounded by $1/\sqrt{c}$. This ensures that the embeddings remain valid within the Poincaré ball throughout the dilation process.

Finally, the rectified embeddings are projected back to Euclidean space for subsequent processing:

$$\mathbf{h}_i^{out} = \text{LayerNorm}(\text{Log}_{\mathbf{0},\varepsilon}^c(\mathbf{z}_i^{(2)})). \quad (16)$$

## 5. Experiments

### 5.1. Implementation Details

In the proposed CHR, the manifold curvature parameter $c$ is initialized to $1.0$, while the hierarchy scaling hyperparameters are set to $\alpha = 1.0$ and $\gamma = 2.0$. The numerical

stability margin is fixed at $\varepsilon = 10^{-5}$. The boundary edge regions radii set as R=0.02. The model is optimized using the AdamW optimizer (Loshchilov & Hutter, 2019) coupled with a cosine annealing learning rate scheduler (Loshchilov & Hutter, 2017). To stabilize the early training dynamics, we incorporate an initial linear warmup phase. The CamPoint (Zhang et al., 2025a) backbone architecture is adopted as the foundational feature encoder, with the CHR seamlessly integrated after major feature extraction stage. Crucially, to ensure a strictly fair comparison with baseline methods, we report performance metrics based on standard single-view evaluation, foregoing any multi-view voting mechanisms during inference. All computational experiments were conducted on a workstation equipped with a single NVIDIA GeForce RTX 4090 GPU.

## 5.2. Main Results

**Semantic Segmentation on S3DIS Dataset.** The S3DIS (Armeni et al., 2016) dataset is a widely adopted benchmark for indoor scene understanding. It comprises six large-scale areas with annotated 3D point clouds. Standard evaluation protocols are followed, with results on Area 5 and 6-fold cross-validation reported. The mean Intersection over Union (mIoU) is utilised to quantify segmentation performance.

As illustrated in Table 1, PointCHR achieves a new state-of-the-art performance, attaining an mIoU of 86.0% on Area 5 and 89.1% on the 6-fold cross-validation. This represents significant improvements over prior leading methods, including CamPoint and Sonata, demonstrating the effectiveness of our geometry-aware hyperbolic rectification approach in enhancing semantic segmentation capabilities.

We utilize edge-strip segmentation as a representative stress test. Boundary zones serve as the ideal proxy because they represent the intersection of peak geometric complexity and semantic ambiguity. As shown in Table 2, PointCHR achieves consistent improvements in these edge regions. Furthermore, the holistic SOTA performance reported in Table 1 confirms that our method generalizes well beyond just boundaries, effectively rectifying the high-curvature embeddings present in the interior as well.

Figure 2 (b) and (c) dissects the impact of geometric complexity on segmentation performance. The background histograms reveal a severe long-tail distribution of curvature, where geometrically complex points are extremely sparse. As shown in Figure 2(b), the Euclidean baseline suffers from a performance collapse in this high-curvature tail due to data scarcity. In contrast, Figure 2(c) demonstrates the efficacy of our PointCHR. Upon introducing the geometry-aware hyperbolic rectification, the performance curve in the tail region is significantly flattened and uplifted. Notably, PointCHR maintains robust mIoU even in the extreme tail, where the baseline fails. This improvement is consistent

*Table 1.* Semantic segmentation mIoU results (%) on the S3DIS dataset are evaluated on Area5 and 6-fold cross-validation.

| METHOD | REF. | AREA5 | 6-FOLD |
|---|---|---|---|
| PTV3 | CVPR'24 | 73.4 | 77.7 |
| HPENET | AAAI'24 | 72.7 | 78.7 |
| PDNET-XXL | AAAI'24 | 72.3 | 78.3 |
| PCM | AAAI'25 | 74.1 | / |
| DEEPLA-120 | CVPR'25 | 75.7 | 79.8 |
| CAMPOINT | CVPR'25 | 83.3 | / |
| SONATA | CVPR'25 | 76.0 | 82.3 |
| VDG | ICCV'25 | 71.5 | 73.2 |
| POINT-PQAE | ICCV'25 | 61.4 | / |
| POINTCHR | / | **86.0** | **89.1** |

*Table 2.* Quantitative comparison of edge-strip segmentation with baseline on S3DIS, reporting mIoU (%) and F1-scores (%) under different neighborhood radii (R).

| METHOD | OVERALL mIOU | R=0.01 mIOU | F1 | R=0.02 mIOU | F1 | R=0.04 mIOU | F1 |
|---|---|---|---|---|---|---|---|
| CAMPOINT | 83.3 | 74.9 | 84.7 | 79.2 | 89.2 | 81.8 | 92.4 |
| POINTCHR | **86.0** | **75.3** | **86.0** | **80.4** | **90.6** | **84.1** | **93.5** |

across boundary regions and complex interior structures. We attribute this robustness to the exponential expansion capacity of hyperbolic space, which allows our model to embed sparse, high-curvature features with sufficient margin, compensating for the data scarcity in the long-tail distribution. We have observed that our model achieves more significant performance improvements for high-curvature components than for medium-curvature components. We have analyzed this phenomenon in the **supplementary materials**. The visualization in Figure 3 illustrates the qualitative improvements achieved by PointCHR.

**Part Segmentation on ShapeNetPart Dataset.** The ShapeNetPart (Yi et al., 2016) dataset is a widely used benchmark for 3D part segmentation, consisting of 16 object categories with annotated part labels. We follow the standard evaluation protocol, reporting the mIoU for both class-level and instance-level segmentation. As shown in Table 3, the mIoU (%) for all classes (Cls.) and instances (Ins.) are reported. PointCHR attains SOTA results, achieving an instance mIoU of 87.0% and a class mIoU of 85.7%. This indicates that PointCHR effectively captures fine-grained geometric subtleties, particularly at part transition regions. These transition zones are geometrically characterized by high curvature and discontinuity, representing the hard samples where Euclidean methods often suffer from feature over-smoothing. By leveraging the exponential expansion of hyperbolic space, our method preserves the distinct manifold structures of adjacent parts, ensuring precise boundary delineation even in objects with complex topology.

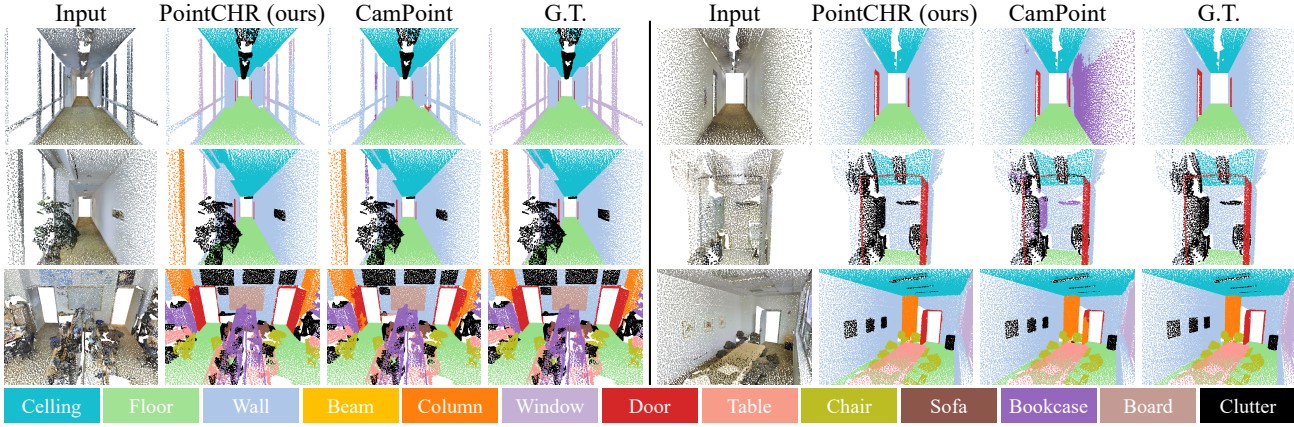

*Figure 3.* Qualitative comparison of segmentation results on the S3DIS Area5 dataset. PointCHR effectively preserves fine-grained boundary details and semantic consistency, significantly outperforming the baseline in challenging scenarios.

*Table 3.* Part segmentation results on ShapeNetPart.

| METHOD | REF. | INS. | CLS. |
|---|---|---|---|
| MAMBA3D | ACMM 24 | 85.7 | 83.6 |
| PCM | AAAI 25 | 86.9 | 85.0 |
| POINT-MODE | IJCAI 25 | 86.5 | 84.6 |
| SI-MAMBA | CVPR 25 | 85.9 | / |
| POINTSD | ICCV 25 | 86.1 | 84.5 |
| POINT-PQAE | ICCV 25 | 86.1 | 84.6 |
| ZIGZAGPOINTMAMBA | NEURIPS 25 | 85.8 | 84.2 |
| POINTPETL | ICML 25 | 86.2 | 84.7 |
| POINTCHR | | **87.0** | **85.7** |

*Table 4.* Classification OA (%) and mAcc(%) on ModelNet40.

| METHOD | REF. | OA | MACC |
|---|---|---|---|
| LINNET | NEURIPS 24 | 93.6 | 91.0 |
| PCM | AAAI 25 | 93.4 | 90.7 |
| CAMPOINT | CVPR 25 | 93.6 | 91.3 |
| POINT-MODE | IJCAI 25 | 93.6 | / |
| SI-MAMBA | CVPR 25 | 92.7 | / |
| POINTSD | ICCV 25 | 93.7 | / |
| POINT-PQAE | ICCV 25 | 92.8 | / |
| ZIGZAGPOINTMAMBA | NEURIPS 25 | 93.2 | / |
| POINTCHR | | **93.7** | **92.0** |

**Shape Classification on ScanObjectNN and ModelNet40 Dataset.** To comprehensively evaluate the generalization capability of PointCHR, we conduct experiments on two distinct benchmarks: ModelNet40 (Wu et al., 2015), which provides synthetic CAD models for assessing representation capability on canonical shapes, and ScanObjectNN (Uy et al., 2019), a real-world dataset containing noisy and occluded scans to verify robustness against geometric imperfections. In contrast to segmentation tasks which require dense point-wise discrimination, classification relies on capturing the global structural skeleton of objects. In this context, high-curvature points serve as the most informative structural anchors for distinguishing similar categories, making our curvature-aware approach inherently suitable for this task.

We initiate our evaluation on the synthetic ModelNet40 dataset, with results summarized in Table 4. PointCHR achieves an Overall Accuracy (OA) of 93.7% and a remarkable Mean Accuracy (mAcc) of 92.0%. The superior mAcc demonstrates that our method maintains high discriminative power on tail classes, particularly those characterized by intricate geometric structures or limited training samples.

This result confirms that explicitly modeling the long-tail curvature distribution enhances the discriminative power of global shape descriptors, setting a strong foundation for handling more complex data.

Proceeding to the more challenging real-world domain, we report results on hardest ScanObjectNN variant, PB_T50_RS, a dataset characterized by severe background clutter and occlusion. As shown in Table 5, PointCHR achieves a SOTA OA of 92.7% and mAcc of 91.7%. This significant improvement suggests that the proposed CHR is highly robust to geometric noise. By expanding the embedding space for high-curvature features, PointCHR effectively distinguishes intrinsic structural details from background noise, validating its efficacy in real-world scenarios.

### 5.3. Ablation Study

We conduct a component-wise ablation study to evaluate the individual and joint contributions of the core in PointCHR: HST and PCP. Results are reported on ScanObjectNN and S3DIS Area5, as summarized in Table 6.

Table 5. Classification OA (%) and mAcc(%) on ScanObjectNN. The ScanObjectNN results are reported on the PB_T50_RS variant.

| METHOD | REF. | OA | MACC |
|---|---|---|---|
| LINNET | NEURIPS 24 | 88.2 | 86.6 |
| KPCONVX-L | CVPR 24 | 89.3 | 88.1 |
| MAMBA3D | ACMM 24 | 91.8 | / |
| PCM | AAAI 25 | 88.1 | 86.6 |
| CAMPOINT | CVPR 25 | 92.1 | 91.1 |
| SI-MAMBA | CVPR 25 | 87.3 | / |
| POINTSD | ICCV 25 | 90.1 | / |
| ZIGZAGPOINTMAMBA | NEURIPS 25 | 88.7 | / |
| POINTPETL | ICML 25 | 90.1 | / |
| POINTCHR | | **92.7** | **91.7** |

Table 6. The efficacy of each component in PointCHR.

| BASELINE | CHR | | SCANOBJECTNN | S3DIS |
|---|---|---|---|---|
| | PCP | HST | OA(%) | MIOU(%) |
| ✓ | | | 92.1 | 83.3 |
| ✓ | | ✓ | 92.2 | 84.1 |
| ✓ | ✓ | | 92.5 | 84.8 |
| ✓ | ✓ | ✓ | **92.8** | **86.0** |

Table 7. Generalization analysis of the proposed CHR.

| METHOD | SCANOBJECTNN OA(%) | S3DIS MIOU(%) |
|---|---|---|
| POINTMLP | 85.4 | / |
| POINTMLP+CHR | 86.2(**+0.8**) | / |
| DELA | 90.4 | 73.2 |
| DELA+CHR | 90.9(**+0.5**) | 75.3(**+2.1**) |
| POINTNEXT-S | 87.7 | 63.4 |
| POINTNEXT-S+CHR | 88.3(**+0.6**) | 65.3(**+1.9**) |

Table 8. Comparison of segmentation performance, parameter size, computational cost and inference latency on S3DIS Area 5.

| METHOD | MIOU (%) | PARAMS. (M) | FLOPS (G) | LATENCY (MS) |
|---|---|---|---|---|
| POINTNEXT-XL | 70.8 | 46.1 | 84.8 | / |
| POINTVECTOR-XL | 72.3 | 24.1 | 58.5 | / |
| PDNET-XXL | 72.3 | 35.6 | 12.0 | / |
| PTV3 | 73.4 | 46.2 | 2.7 | 94 |
| DEEPLA-120 | 75.7 | 30.3 | 24.7 | / |
| CAMPOINT | 83.3 | 15.8 | 3.7 | 22 |
| POINTCHR | 86.0 | 21.0 | 4.0 | 25 |

Starting from the baseline model, we observe that incorporating HST yields a consistent performance improvement. This suggests that modeling semantic interactions in hyperbolic space helps enhance feature discriminability, especially for structurally complex regions. Similarly, enabling PCP alone leads to a more pronounced gain on both benchmarks highlighting the importance of curvature-guided capacity reallocation for capturing fine-grained geometric details.

Notably, combining HST and PCP achieves the best overall performance across both tasks. The complementary gains indicate that HST and PCP address different aspects of the representation challenge: while HST improves semantic mixing under hyperbolic geometry, PCP explicitly redistributes representational capacity according to local geometric complexity. Their synergy effectively alleviates representation crowding and leads to more robust and discriminative point cloud representations.

### 5.4. Generalization Analysis

To verify the generalization capability and plug-and-play nature of our proposed method, we integrate the CHR into three representative backbones: PointMLP, DeLA, and PointNext-s. As summarized in Table 7, the integration of CHR yields consistent performance gains across diverse architectures and tasks. Specifically, on the ScanObjectNN classification benchmark, all backbones exhibit steady improvements in OA. Notably, on the more challenging S3DIS semantic segmentation task, the benefits are even more pronounced, with DeLA and PointNext-s achieving significant

mIoU increases of 2.1% and 1.9%, respectively. These results demonstrate that CHR is model-agnostic and effectively enhances the feature discriminability of existing methods without requiring architectural modifications.

### 5.5. Model Complexity

We evaluate the trade-off between segmentation accuracy and resource consumption in Table 8. PointCHR outperforms previous vanilla methods, achieving a dominant mIoU of 86.0%. Critically, this top-tier performance is achieved with a lean architecture of only 21.0M parameters. Compared with PointNeXt-XL and PTV3, which rely on scaling up parameters to enhance capacity, PointCHR proves that utilizing curvature-aware hyperbolic embeddings is a more parameter-efficient strategy for capturing fine-grained semantics. Consequently, the proposed model surpasses the previous SOTA, CamPoint, by 2.7% in mIoU while adding FLOPs from 3.7G to 4.0G. This result highlights the efficacy of our design: by rectifying feature crowding in high-curvature regions, we achieve maximizing representational capacity within a constrained parameter budget.

### 5.6. Curvature-stratified Quantitative Analysis

To examine PointCHR under different geometric regimes, we conduct a curvature-stratified evaluation on S3DIS. Points are partitioned into four curvature intervals, and segmentation performance is reported for each bin. We also report the point ratio of each bin and the learned scaling factors for boundary and non-boundary regions.

*Table 9.* Curvature-stratified quantitative results on S3DIS. PointCHR consistently improves segmentation performance across all curvature regimes, with the most substantial gain observed in the highest-curvature bin. B and nB denote boundary and non-boundary regions.

| Curvature Bin | Point Ratio (%) | $s$ (B / nB) | Baseline mIoU (%) | PointCHR mIoU (%) | $\Delta$ |
|---|---|---|---|---|---|
| $[0.00, 0.05)$ | 83.64 | 1.22 / 1.13 | 84.39 | 86.59 | +2.20 |
| $[0.05, 0.10)$ | 10.34 | 1.84 / 1.43 | 76.75 | 80.91 | +4.16 |
| $[0.10, 0.15)$ | 3.89 | 2.13 / 1.61 | 73.05 | 76.78 | +3.73 |
| $[0.15, 0.20]$ | 2.13 | 2.13 / 1.76 | 66.22 | 76.62 | +10.40 |

*Table 10.* Distribution of the learned scaling factor $s$ across curvature bins and spatial regions. The threshold $s = 1.13$ is used as a data-driven reference value, corresponding to the mean scaling factor of low-curvature non-boundary points.

| Curvature Bin | Point Ratio (%) | Avg. $s$ (nB) | $(s > 1.13)$ nB Ratio (%) | Avg. $s$ (B) | $(s > 1.13)$ B Ratio (%) |
|---|---|---|---|---|---|
| $[0.00, 0.05)$ | 83.64 | 1.13 | 19.81 | 1.22 | 24.18 |
| $[0.05, 0.10)$ | 10.34 | 1.43 | 40.64 | 1.84 | 53.12 |
| $[0.10, 0.15)$ | 3.89 | 1.61 | 49.47 | 2.13 | 61.99 |
| $[0.15, 0.20]$ | 2.13 | 1.76 | 55.50 | 2.13 | 63.00 |

As shown in Table 9, PointCHR consistently improves performance across all curvature bins, with gains increasing from +2.20% in the lowest-curvature bin to +10.40% in the highest-curvature bin. Although the highest-curvature points account for only 2.13% of all points, PointCHR improves their mIoU from 66.22% to 76.62%, demonstrating its effectiveness in mitigating Euclidean representation degradation in geometrically complex regions.

### 5.7. Scaling Factor and Boundary Distribution

We also analyze the learned scaling factor $s$ across curvature bins and spatial regions. Since $s$ controls the strength of feature rectification, its distribution reflects whether PointCHR learns an adaptive, geometry-aware transformation instead of applying uniform scaling. This trend is also consistent with the qualitative observations in Figure 2 (c) and Figure 3. Table 10 shows a clear positive correlation between local curvature and the learned scaling factor. As curvature increases, both the average $s$ and the proportion of high-scaling points consistently increase.

These results demonstrate that PointCHR assigns stronger rectification to geometrically complex regions. Moreover, boundary points consistently receive larger scaling factors than non-boundary points within the same curvature range, indicating that the learned transformation is both curvature-aware and structure-aware. This provides mechanistic evidence that PointCHR improves performance by selectively enhancing high-curvature and structurally critical regions, rather than relying on uniform feature expansion.

### 5.8. Quantitative Analysis of Representation Crowding

To verify whether PointCHR alleviates Representation Crowding, we evaluate two complementary metrics on

*Table 11.* Quantitative verification of representation crowding on S3DIS. Higher values indicate less representation crowding. B and P denote Base and PointCHR, respectively.

| Curvature Bin | ED-B | ED-P | M-B | M-P |
|---|---|---|---|---|
| $[0.00, 0.05)$ | 7.31 | 6.38 | -2.62 | **-0.21** |
| $[0.05, 0.10)$ | 10.23 | 9.86 | -6.12 | **-0.97** |
| $[0.10, 0.15)$ | 9.06 | **10.05** | -5.14 | **-0.89** |
| $[0.15, 0.20]$ | 7.49 | **8.99** | -3.15 | **-0.64** |

S3DIS: Effective Dimensionality (ED) for intrinsic representational capacity and classification margin (M) for class separability. As shown in Table 11, PointCHR consistently improves the classification margin across all curvature bins, indicating clearer class boundaries. In the high-curvature bins $[0.10, 0.20]$, PointCHR also increases ED, suggesting enhanced representational capacity in geometrically complex regions. These results confirm that PointCHR effectively alleviates representation crowding by improving both feature capacity and class separability.

## 6. Conclusion

This study introduces PointCHR, a novel curvature-aware hyperbolic rectification designed to enhance point cloud representation learning. By exploiting the exponential expansion properties of hyperbolic manifolds to overcome the capacity limitations of Euclidean space, PointCHR effectively addresses the representation crowding dilemma, exhibiting exceptional discriminative capability in high-curvature geometric details. Extensive experiments across multiple benchmarks validate the superiority of PointCHR, achieving SOTA performance in semantic segmentation, part segmentation, and shape classification tasks.

## Acknowledgements

This work was supported by the Science and Technology Development Fund of Macau Project 0096/2023/RIA2, 0123/2022/A3, 0044/2024/AGJ, 0140/2024/AGJ, the Chinese National Natural Science Foundation Projects 62406320, and the AI Supercomputing Center of Macau University of Science and Technology (MUST). We also acknowledge the technical support provided by the Information Technology Development Office of MUST, as well as the reviewers AVjy, e3C6, v2GW, and 1Vun for their comments and suggestions on the manuscript.

## Impact Statement

This paper presents work whose goal is to advance the field of Machine Learning. There are many potential societal consequences of our work, none which we feel must be specifically highlighted here.

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

## A. Mechanistic Analysis of Performance Disparities Between Medium and High Curvature Parts.

As demonstrated in manuscript Figure 2 (c), an intriguing phenomenon is observed, in which the performance enhancement in the medium-curvature regions is less pronounced in comparison to the high-curvature regions, resulting in a distinct trend of tail-rising. This disparity is attributed to geometric characteristics. In contrast to high-curvature parts, which are characterised by distinct topological invariants, medium-curvature parts typically manifest as smooth transitional surfaces, exhibiting high intrinsic ambiguity. This feature uncertainty imposes an implicit radial constraint during the embedding process, predisposing medium-curvature samples to distribute within the intermediate region of the Poincaré ball rather than approaching the extreme boundary. While this region offers certain advantages in terms of negative curvature over Euclidean space, thereby yielding moderate performance gains, it lacks the exponential growth of the conformal factor found near the boundary. Consequently, medium-curvature parts are unable to access the explosive volume expansion and superior feature separability available at the manifold limit. The non-linearity of geometric benefits ultimately leads to significantly smaller performance margins for medium-curvature parts compared to their high-curvature counterparts, which occupy the high-gain boundary regions.

Crucially, this radial distribution bias translates directly into challenges within the gradient dynamics. Despite the implementation of the AdamW optimiser to address gradient scaling issues through adaptive moment estimation, the optimization dynamics remain predominantly constrained by the underlying Riemannian geometry. In the Poincaré ball model, the conformal factor of the metric tensor grows exponentially with radius. This geometric property has the capacity to inherently reshape the loss landscape. In the vicinity of the boundary (high-curvature regions), the hyperbolic distance exhibits hyper-sensitivity to parameter perturbations, giving rise to a surface of high steepness that provides strong geometric supervision signals. Conversely, in the near-origin region occupied by medium-curvature parts, the conformal factor varies slowly, resulting in a locally flat optimization landscape. Despite AdamW's endeavours to standardise update procedures, this intrinsic deficiency in optimisation dynamics, attributable to the manifold geometry, remains unaddressed. The flatness that is associated with medium-curvature parts leads to geometric gradients with insufficient discriminative power, causing the model to converge to suboptimal solutions.

## B. Limited Gains on Classification.

The main advantage of PointCHR appears when local geometric complexity is important. Datasets such as ModelNet40 are dominated by global shape recognition. Coarse structural cues are sufficient and performance is already saturated. For ModelNet40, prior work has noted that these benchmarks are already in a high-performance regime, where further gains are typically small and increasingly difficult to obtain (Van den Herrewegen et al., 2023). Therefore, improvements are less likely to appear uniformly in global averages, and more likely to emerge in geometrically complex or difficult regions, which is consistent with our fine-grained analysis. In contrast, PointCHR is designed to alleviate representation crowding in high curvature regions. Therefore, its advantages are more pronounced in tasks that require fine-grained local geometric discrimination. This trend is reflected in the experiments. On S3DIS, where accurate local geometry modeling is essential, PointCHR achieves more improvements. This indicates that curvature aware hyperbolic rectification is effective for precise local feature separation. On ModelNet40, PointCHR improves mAcc to 92.0%, suggesting enhanced discrimination for difficult categories. On ScanObjectNN, which contains noise, occlusion, and irregular sampling, PointCHR achieves 92.7% OA and 91.7% mAcc, surpassing recent methods. Although gains on clean global classification tasks are moderate, the consistent improvements across real scan classification and scene understanding demonstrate that PointCHR is effective when local geometric complexity is the dominant factor.

## C. Necessity of Hyperbolic Modeling

To examine whether the improvement comes merely from anisotropic feature scaling or from the hyperbolic formulation itself, we compare PointCHR with two Euclidean variants. The first is an Anisotropic Euclidean Scaling (AES) variant, which performs curvature-aware scaling directly in Euclidean space. The second is a CurvNet-style Euclidean variant, which applies PCP-derived weights without hyperbolic mapping.

As shown in Table 12, AES improves upon the baseline, suggesting that curvature-aware anisotropic scaling is helpful. However, it remains inferior to the hyperbolic formulation. This gap indicates a fundamental difference in representational capacity. In Euclidean space, the available volume grows only polynomially with radius, so anisotropic scaling can only provide limited relief for representation crowding. In contrast, hyperbolic space provides exponential volume growth toward

the boundary, allowing PointCHR to allocate larger representational capacity to high-curvature regions while reducing interference from simpler regions. Therefore, AES serves as a useful Euclidean approximation, but the hyperbolic manifold is essential for resolving representation crowding in a principled and capacity-aware manner.

*Table 12.* Comparison with Euclidean variants on S3DIS and ScanObjectNN. AES denotes anisotropic Euclidean scaling, while Cur denotes the CurvNet-style Euclidean variant. The hyperbolic formulation achieves the best performance, indicating that the gain does not come from feature scaling alone.

| Method | S3DIS mIoU (%) | ScanObjectNN OA (%) |
|---|---|---|
| Baseline | 83.30 | 92.10 |
| Euclidean (AES) | 84.93 | 92.35 |
| Euclidean (Cur) | OOM | OOM |
| Hyperbolic (PointCHR) | **86.01** | **92.70** |

## D. Verification of Representation Crowding

Prior work on hyperbolic representation learning shows that Euclidean embeddings are limited in representing hierarchical or highly branching structures. Poincaré embeddings demonstrate that hyperbolic geometry can compactly capture hierarchy and similarity (Nickel & Kiela, 2017), while hyperbolic space provides an exponential advantage in representational capacity over Euclidean space due to its exponential volume growth (Sala et al., 2018). Atigh et al.(Atigh et al., 2022) show that image segmentation exhibits latent hierarchical structure and benefits from hyperbolic modeling, especially in low-dimensional embeddings, suggesting that real-world visual data may not be efficiently represented in Euclidean. $H^2$ARN indicate that embedding such hierarchical structures in Euclidean or standard Riemannian spaces can lead to a crowding effect, where semantically distinct entities become harder to separate due to limited representational capacity(Li et al., 2026).

Extending these insights to 3D geometry, we measure the Gromov $\delta$-hyperbolicity of point cloud manifolds to be as low as 0.02, indicating a near-tree topology that Euclidean embeddings struggle to accommodate. Motivated by these observations, we define representation crowding as a relative phenomenon where Euclidean embeddings tend to compress geometrically complex, high-curvature regions into lower-capacity subspaces, leading to reduced representational diversity and increased class overlap. In this sense, crowding is not an absolute property of a feature space, but is measured relative to a more geometry-aware representation. We quantify this phenomenon using two complementary metrics on S3DIS dataset: Effective Dimensionality (ED), which reflects intrinsic representational capacity, as shown in Table 13, and classification margin, which measures class separability, as shown in Table 14. Together, these two metrics characterize the two key aspects of representation crowding: capacity and separability. These results indicate that PointCHR mitigates representation crowding, especially in high-curvature regions. In the more complex bins [0.10,0.20], PointCHR increases ED and margin, indicating both enhanced representational capacity and clearer class boundaries. This shows that PointCHR does not merely reshape the feature distribution, but effectively alleviates the low-capacity compression and boundary entanglement that characterize crowded Euclidean representations.

## E. Comparison with Resampling Baseline

Curvature guided resampling (CGR) is a data level baseline for handling sparsity in high curvature regions. We implement this baseline on S3DIS by using the learned weights in Eq.(12) to prioritize high curvature points.

The Table 15 results show that curvature guided resampling underperforms both the baseline and PointCHR. This indicates that representation crowding cannot be resolved at the data level. Resampling modifies the original point distribution and disrupts spatial continuity and local topology, which degrades the ability of the backbone to model consistent geometric dependencies. Increasing point density in high curvature regions does not improve the intrinsic capacity of the feature space and therefore cannot resolve feature compression. In contrast, PointCHR operates in the embedding space and leverages the exponential radial capacity of hyperbolic geometry to perform curvature-based nonlinear feature expansion. This mechanism allocates more representational capacity to geometrically complex regions without modifying spatial structure or discarding contextual information. The results suggest that the limitation lies in feature space capacity rather than point density. A manifold based transformation is therefore more suitable than data level resampling.

*Table 13.* Curvature-stratified ED analysis. ED characterizes the intrinsic capacity of representations under different curvature levels, serving as a capacity-oriented measure of representation crowding.

| Curvature Bin | Base | AES | Our |
|---|---|---|---|
| $[0.00, 0.05)$ | 7.31 | 7.25 | 6.38 |
| $[0.05, 0.10)$ | 10.23 | 10.48 | 9.86 |
| $[0.10, 0.15)$ | 9.06 | 9.23 | 10.05 |
| $[0.15, 0.20]$ | 7.49 | 7.93 | 8.99 |

*Table 14.* Curvature-stratified margin analysis. Classification margin characterizes class separability under different curvature levels, serving as a separability-oriented measure of representation crowding.

| Curvature Bin | Base | AES | Our |
|---|---|---|---|
| $[0.00, 0.05)$ | -2.62 | -2.32 | -0.21 |
| $[0.05, 0.10)$ | -6.12 | -5.97 | -0.97 |
| $[0.10, 0.15)$ | -5.14 | -5.01 | -0.89 |
| $[0.15, 0.20]$ | -3.15 | -2.87 | -0.64 |

*Table 15.* Comparison of segmentation performance on S3DIS under different resample method.

| Method | S3DIS mIoU (%) |
|---|---|
| Baseline | 83.30 |
| CGR | 79.54 |
| PointCHR | 86.01 |

# F. Sensitivity and Robustness Analysis

We conduct additional analyses on key hyperparameters, curvature estimation strategies, noise interference, and sparse point cloud conditions to evaluate the sensitivity and robustness of PointCHR. Specifically, the experiments cover neighborhood size $K$, curvature parameter $c$, hyperbolic coefficients, curvature estimation methods, curvature noise, missing curvature, and test-time point density reduction.

## F.1. Sensitivity to Neighborhood Size and Curvature Parameter

As shown in Table 16, PointCHR remains stable across different neighborhood sizes, indicating low sensitivity to the KNN scale $K$. It also maintains stable performance under different manifold curvature parameters $c$.

*Table 16.* Sensitivity analysis of neighborhood size $K$ and curvature parameter $c$ on S3DIS. PointCHR remains stable across different KNN scales and manifold curvature values.

| $K$ | S3DIS mIoU (%) | $c$ | S3DIS mIoU (%) |
|---|---|---|---|
| 16 | 85.98 | 0.5 | 85.87 |
| 24 | 86.01 | 1.0 | 86.01 |
| 32 | 85.97 | 2.0 | 85.93 |

## F.2. Sensitivity to Hyperbolic Coefficients

As reported in Table 17, PointCHR achieves strong performance under a broad range of hyperbolic coefficient combinations of $\alpha$ and $\gamma$, suggesting that the method is robust to the scaling intensity.

*Table 17.* Sensitivity analysis of hyperbolic coefficients on S3DIS. Rows denote $\alpha$, columns denote $\gamma$, and each entry reports S3DIS mIoU (%).

| $\alpha\backslash\gamma$ | 1 | 2 | 3 |
|---|---|---|---|
| 1 | 85.88 | 86.01 | 85.06 |
| 2 | 85.58 | 85.66 | 86.04 |
| 3 | 85.42 | 85.56 | 85.00 |

*Table 18.* Comparison of curvature estimation strategies. PCA-based curvature estimation achieves competitive or better performance with lower computational cost than MLS.

| Dataset | Method | mIoU / OA | Time | Epochs |
|---|---|---|---|---|
| S3DIS | MLS | 85.97 | 16H | 150 |
| S3DIS | PCA | 86.01 | 10H | 150 |
| ScanObjectNN | MLS | 92.64 | 54H | 400 |
| ScanObjectNN | PCA | 92.71 | 34H | 400 |

### F.3. Comparison of Curvature Estimation Strategies

We compare the PCA-based curvature estimation strategy with the Moving Least Squares (MLS) method in Table 18. PCA achieves slightly better performance with significantly lower computational cost. In contrast, higher-order fitting with MLS does not provide additional benefits and introduces larger overhead. This supports the efficiency and lightweight design of the PCA-based curvature estimation strategy.

### F.4. Robustness to Noisy and Sparse Point Clouds

Large-scale point clouds often involve noise, occlusion, and nonuniform sampling, which may lead to unstable curvature estimation and reduced point density. To evaluate the robustness of PointCHR under such conditions, we conduct controlled perturbation experiments on S3DIS, as summarized in Table 19. Specifically, we consider three types of test-time perturbations without retraining: curvature noise, missing curvature, and point density reduction. For curvature noise, Gaussian noise is injected into curvature values to simulate noisy curvature estimation. For missing curvature, a certain ratio of curvature values is randomly set to zero. For point density reduction, random point masking is applied at test time to simulate sparse point observations.

The results in Table 19 show that PointCHR remains stable under noisy and sparse point cloud conditions. Under curvature perturbation, the performance remains above 85% mIoU for moderate noise levels and only drops by 1.23% when $\sigma = 0.20$. Under missing curvature, PointCHR maintains 85.10% mIoU even when 30% of curvature values are removed. This indicates that the method does not rely on perfectly estimated curvature values. Instead, curvature serves as a modulation signal, while the backbone can absorb local estimation errors through contextual aggregation.

For point density variation, the performance remains nearly unchanged up to a masking ratio of 30%, with only a 0.05% drop. Noticeable degradation appears only under severe masking ratios of 40%–50%. These results demonstrate that PointCHR preserves discriminative geometric structures under reduced point density and is robust to curvature noise, curvature sparsity, and test-time point masking.

## G. Implementation Details and Hyperparameter Settings

Our implementation follows the same basic training protocol as CamPoint. The key training hyperparameters are summarized in Table 20. For all datasets, we use cosine decay as the learning rate scheduler.

During training, we apply standard data augmentations, including random rotation, random scaling within $[0.8, 1.2]$, and random point dropout with a ratio of 10%, to improve robustness.

*Table 19.* Robustness analysis on S3DIS under curvature noise, missing curvature, and point density variation. All models are trained on clean data and directly evaluated under perturbed test-time conditions without retraining.

| Setting | Perturbation | mIoU (%) | $\Delta$ |
|---|---|---|---|
| | Clean | 86.01 | – |
| | $\sigma = 0.01$ | 86.37 | +0.36 |
| Curvature noise | $\sigma = 0.05$ | 85.58 | -0.43 |
| | $\sigma = 0.10$ | 85.49 | -0.52 |
| | $\sigma = 0.20$ | 84.78 | -1.23 |
| | Clean | 86.01 | – |
| | 10% | 85.84 | -0.17 |
| Missing curvature | 20% | 85.61 | -0.40 |
| | 30% | 85.10 | -0.91 |
| | Clean | 86.01 | – |
| | 5% | 86.43 | +0.32 |
| | 10% | 86.00 | -0.01 |
| Point masking | 20% | 86.03 | +0.02 |
| | 30% | 85.96 | -0.05 |
| | 40% | 84.97 | -1.04 |
| | 50% | 83.51 | -2.50 |

*Table 20.* Training hyperparameter settings across different datasets.

| Dataset | Batch Size | Epochs | LR | Scheduler | Warmup |
|---|---|---|---|---|---|
| S3DIS | 8 | 150 | $1e-3$ | Cosine decay | 10 |
| ShapeNetPart | 32 | 400 | $2e-3$ | Cosine decay | 30 |
| ScanObjectNN | 32 | 400 | $5e-4$ | Cosine decay | 30 |
| ModelNet40 | 32 | 400 | $5e-4$ | Cosine decay | 30 |

To map the raw curvature $\kappa$ into a stable numerical range, we apply min–max normalization followed by clipping:

$$\hat{\kappa} = \text{clamp}\left(\frac{\kappa - \kappa_{\min}}{\kappa_{\max} - \kappa_{\min}}, 0, 1\right).$$

In practice, we set $\kappa_{\max} = 0.2$ as a threshold to suppress extreme outliers, ensuring stable estimation of the scaling factor $s_i$.

