# OpenReview forum: "PointCHR: Point Cloud Analysis via Curvature-Aware Hyperbolic Rectification"
_ICML.cc/2026/Conference — ICML 2026 regular_

### Official Review · Reviewer_AVjy · 2026-03-02

**Soundness:** 3
**Presentation:** 3
**Significance:** 3
**Originality:** 3
**Overall Recommendation:** 4
**Confidence:** 4

**Summary:**

This paper proposes PointCHR, a curvature-aware hyperbolic feature renormalization framework for 3D point cloud analysis. The method leverages the exponential volume growth of hyperbolic space to align local geometric complexity with embedding capacity, aiming to alleviate representation crowding in Euclidean space.

The framework includes three components: a hyperbolic space transformation (HST), a point-wise curvature prior (PCP) for radial scaling, and a closed-form geodesic dilation (CGD) to stabilize scaling operations.

Experiments on S3DIS, ShapeNetPart, ModelNet40, and ScanObjectNN show consistent improvements, particularly in semantic segmentation. Conceptually, the approach allocates embedding capacity adaptively by mapping high-curvature points toward regions with larger hyperbolic capacity while keeping low-curvature points in more stable regions.

**Compliance With Llm Reviewing Policy:**

Affirmed.

**Final Justification:**

I would like to thank the authors for their detailed and comprehensive rebuttal. It has addressed the majority of my initial concerns, particularly those regarding mathematical consistency, hyperparameter sensitivity, and scaling behavior.

However, despite these positive additions, I still have two minor reservations:

1. **Robustness in Real-World Outdoor Scenes:** While the perturbation-based simulations on S3DIS provide a useful reference, evaluating the method on actual large-scale outdoor LiDAR point cloud data would offer much stronger and more convincing evidence of its robustness in realistic, noisy, and complex environments.
2. **Computational Overhead:** The actual training overhead reported is non-negligible in practical applications and may require further analysis or optimization in the future.

Despite the aforementioned limitations, the core contribution of this pape: curvature-aware hyperbolic rectification holds conceptual value. The rebuttal has alleviated my previous concerns. Therefore, I am revising my final rating to a **Weak Accept**. I strongly recommend that the authors include the newly provided analyses in the appendix and explicitly discuss the computational overhead as well as the limitations regarding outdoor LiDAR applications in the final manuscript.

**Key Questions For Authors:**

- Could the authors provide a formal definition and quantitative analysis of “representation crowding” to more clearly demonstrate how CHR mitigates it?

- How can we attribute the gains specifically to hyperbolic geometry, rather than to additional learnable scaling mechanisms? Comparisons with anisotropic Euclidean scaling or curve-based aggregation methods (e.g., CurveNet) would help clarify whether hyperbolic geometry is essential.

- The manuscript suggests larger improvements in high-curvature regions. Could the authors provide curvature-stratified quantitative results and visualizations to illustrate performance across different geometric regimes?

- How sensitive is the method to inaccuracies in curvature estimation? An analysis of robustness under noisy curvature estimates would further strengthen the claims.

**Limitations:**

- Hyperbolic operations introduce additional computational overhead compared to purely Euclidean formulations.

- The manuscript would benefit from more consistent and clearer notation.

- Performance gains are more evident on segmentation tasks, while improvements on classification benchmarks are relatively modest.

- If the above concerns are satisfactorily addressed, I would be open to revising my score.

**Strengths And Weaknesses:**

### Strengths

- The motivation is clearly articulated: the long-tailed curvature distribution in point clouds may lead to representational imbalance under polynomial volume growth in Euclidean space. The overall reasoning is coherent and self-consistent.

- The work introduces a geometric alignment perspective by matching hyperbolic radial capacity with local geometric complexity, which offers a conceptually novel angle compared to prior hyperbolic approaches focused on hierarchy modeling.

- The closed-form geodesic dilation (CGD) avoids repeated tangent space projections and improves numerical stability.

- Experiments on S3DIS, ShapeNetPart, ModelNet40, and ScanObjectNN demonstrate reasonable empirical scalability.


### Weaknesses

- The notion of “representation crowding” remains largely conceptual. The paper does not provide a precise definition or quantitative validation (e.g., embedding density, margin distribution, entropy, or capacity utilization analysis).

- The necessity of hyperbolic geometry is not fully established. Comparisons with alternative capacity-expansion strategies (e.g., anisotropic Euclidean metrics or learnable metric transformations) would strengthen the argument.

- The manuscript refers to supplementary analysis regarding curvature-dependent improvements, but the corresponding material does not appear to be accessible in the submission system.

- Improvements on classification benchmarks such as ModelNet40 are relatively modest, and the broader generalization implications could be further discussed.

---

> ### Author Rebuttal · Authors · 2026-03-30
>
> *R4Q1*:“representation crowding” is not defined or validated.
>
> **R4A1**:Thank the reviewer for requesting a more rigorous characterization of representation crowding. We define it as the phenomenon in which features from geometrically complex, high-curvature regions are compressed into a low-capacity subspace in Euclidean embeddings, leading to reduced representational diversity and increased class overlap. To validate this effect, we adopt Effective Dimensionality (ED) and Classification Margin on S3DIS dataset.
>
> |Curvature Bin|Base ED ↑|ED ↑|Base Margin ↑|Margin ↑|
> |-|-|-|-|-|
> |[0.00, 0.05)|7.31|6.38|-2.62|-0.21|
> |[0.05, 0.10)|10.23|9.86|-6.12|-0.97|
> |[0.10, 0.15)|9.06|10.05|-5.14|-0.89|
> |[0.15, 0.20]|7.49|8.99|-3.15|-0.64|
>
> These results indicate that PointCHR mitigates representation crowding, especially in high-curvature regions. In the more complex bins [0.10,0.20], PointCHR increases ED and margin, indicating both enhanced representational capacity and clearer class boundaries. This shows that the PointCHR does not merely reshape the feature distribution, but effectively alleviates the low-capacity compression and boundary entanglement that characterize crowded Euclidean representations. We will incorporate this definition and quantitative analysis into the Appendix.
>
> ---
>
> *R4Q2*:The necessity of hyperbolic.
>
> **R4A2**:Thank the reviewer for this question. We compare our method with an Anisotropic Euclidean Scaling (AES) variant and CurveNet variant, which applies the PCP-derived weights but performs feature scaling in Euclidean space.
> |Method|S3DIS(mIoU%)|ScanObjectNN(OA%)|
> |-|-|-|
> |Baseline|83.30|92.10|
> |Euclidean(AES)|84.93|92.35|
> |Euclidean(Cur)|OOM|OOM|
> |Hyperbolic(PointCHR)|**86.01**|**92.70**|
>
> Although AES improves upon the baseline, it remains inferior to the hyperbolic formulation. This gap reflects a fundamental difference in representational capacity: in Euclidean space, the available volume grows only polynomially with radius, so even anisotropic scaling cannot prevent progressive feature crowding as complex points are pushed outward. By contrast, hyperbolic space provides exponential volume growth toward the boundary. Leveraging this property, CHR allocates larger representational capacity to high-curvature features, allowing them to unfold in the embedding space without increasing interference from simpler regions. Therefore, AES offers only limited relief, whereas the hyperbolic manifold is essential for resolving representation crowding in a principled manner.
>
> ---
>
> *R4Q3*: Requests curvature-stratified quantitative results.
>
> **R4A3**:Thank the reviewer. We conduct a curvature-stratified analysis on S3DIS.
>
> |Curvature Bin|Point Ratio (%)|s (Bound / Non-Bound)|Baseline mIoU(%)|PointCHR mIoU(%)|Δ|
> |-|-|-|-|-|-|
> |[0.00, 0.05)|83.64|1.22/1.13|84.39| 86.59|+2.20|
> |[0.05, 0.10)|10.34|1.84/1.43|76.75| 80.91|+4.16|
> |[0.10, 0.15)|3.89|2.13/1.61|73.05| 76.78|+3.73|
> |[0.15, 0.20]|2.13|2.13/1.76|66.22| 76.62|+10.40|
>
> The results exhibit a curvature-dependent trend. As curvature increases, the advantage of PointCHR becomes larger, with the gain rising from +2.20% in low-curvature regions to +10.40% in the highest-curvature bin. This indicates that PointCHR is effective on challenging samples, where Euclidean models suffer the most degradation. Although the highest-curvature points account for only 2.13% of the data, the baseline drops to 66.22%, whereas PointCHR restores it to 76.62%, confirming its ability to alleviate representation crowding precisely where it is most severe.
> This trend is also consistent with the qualitative results in Fig. 2(c) and Fig. 3. These quantitative and qualitative results provide evidence that PointCHR consistently improves performance across different geometric regimes, with the most pronounced benefit in high-curvature regions.
>
> ---
>
> *R4Q4*: Limited gains on classification.
>
> **R4A4**: We appreciate the observation. In **R1A3** and **R2A2**, we analyze the underlying reasons, mainly including the task-dependent distinction between global shape recognition and local geometry modeling, the class-balanced discriminative improvements reflected by mAcc.
>
> ---
>
> *R4Q5*:Missing the supplementary analysis.
>
> **R4A5**:We apologize for the confusion. The curvature-dependent analysis is already included in Page 12 of the manuscript.
>
> ---
>
> *R4Q6*:The model sensitivity.
>
> **R4A6**: The noise perturbation analysis on **R2A4** indicates strong tolerance to curvature estimation errors and supports stability in noisy settings.
>
> ---
>
> *R4Q7*: Introduce additional computational.
>
> **R4A7**:As shown in **R2A3**, although hyperbolic operations introduce additional computation, PointCHR yields a highly favorable accuracy–efficiency trade-off, delivering substantial performance gains with only modest increases in parameters, FLOPs, and latency.
>
> ---
>
> *R4Q8*: Constructive feedback.
>
> **R4A8**:We appreciate the constructive feedback and will incorporate their suggestions into the manuscript.

---

> > ### Author Rebuttal · Reviewer_AVjy · 2026-04-02
> >
> > I appreciate the authors’ efforts in addressing the reviewers’ concerns with additional results and comparisons.
> > The rebuttal addresses several of my main concerns with meaningful additional evidence, especially the curvature-stratified analysis, the Euclidean(AES) comparison, the robustness/sensitivity analysis, and the overhead discussion.
> >
> > That said, some concerns are only partially resolved. In particular, the explanation for the relatively limited gains on classification/part segmentation is still not fully convincing: the rebuttal mainly attributes this to benchmark saturation, but the results in the paper suggest that this point deserves more careful discussion. Likewise, some core claims, such as the strict validation of “representation crowding” and the necessity of the hyperbolic formulation, are still not fully established.
> >
> > Overall, I will consider increasing my score.

---

> > > ### Author Response · Authors · 2026-04-05
> > >
> > > *R4Q1*:More discussion on limited gains
> > >
> > > **R4A1**:For ModelNet40, prior work has noted that these benchmarks are already in a high-performance regime, where further gains are typically small and increasingly difficult to obtain [1,2]. Therefore, improvements are less likely to appear uniformly in global averages, and more likely to emerge in geometrically complex or difficult regions, which is consistent with our fine-grained analysis.
> > >
> > > Indeed, our curvature-stratified results on ShapeNetPart. The results reveal that the improvements brought by PointCHR are not uniform, but instead correlate with geometric complexity. In low-curvature regions, where Euclidean representations are already sufficiently expressive, PointCHR maintains comparable performance. As curvature grows, the larger representational capacity of hyperbolic space enables more effective feature separation, leading to more pronounced improvements.
> > > |Method|Bin 1|Bin 2|Bin 3|Bin 4|Bin 5|
> > > |-|-|-|-|-|-|
> > > |Baseline|0.864|0.829|0.775|0.722|0.679|
> > > |Our|0.922 |0.861|0.829|0.822|0.840|
> > > |Δ|+0.004|+0.021|+0.021|+0.008|+0.039|
> > >
> > > *R4Q2*: Verification of the “representation crowding” and the necessity of hyperbolic。
> > >
> > > **R4A2**:Prior work on hyperbolic representation learning shows that Euclidean embeddings are limited in representing hierarchical or highly branching structures. Poincaré embeddings demonstrate that hyperbolic geometry can compactly capture hierarchy and similarity [3], while hyperbolic space provides an exponential advantage in representational capacity over Euclidean space due to its exponential volume growth [4]. Atigh et al.[5] show that image segmentation exhibits latent hierarchical structure and benefits from hyperbolic modeling, especially in low-dimensional embeddings, suggesting that real-world visual data may not be efficiently represented in Euclidean.  $H^2$ARN indicate that embedding such hierarchical structures in Euclidean or standard Riemannian spaces can lead to a crowding effect, where semantically distinct entities become harder to separate due to limited representational capacity [6].
> > >
> > > Extending these insights to 3D geometry, we measure the Gromov δ-hyperbolicity of point cloud manifolds to be as low as 0.02, indicating a near-tree topology that Euclidean embeddings struggle to accommodate. Motivated by these observations, we define representation crowding as a relative phenomenon, as demonstrated in previous discussions. We quantify it using two complementary metrics: Effective Dimensionality (ED), which reflects intrinsic representational capacity, and classification margin, which measures class separability. They capture the two key aspects of crowding: capacity and separability.
> > >
> > > Curvature-stratified ED
> > > |Bin|Base|AES|Our|
> > > |-|-|-|-|
> > > |1|7.31|7.25|6.38|
> > > |2|10.23|10.48|9.86|
> > > |3|9.06|9.23|10.05|
> > > |4|7.49|7.93|8.99|
> > >
> > > Curvature-stratified Margin
> > > |Bin|Base|AES|Our|
> > > |-|-|-|-|
> > > |1|-2.62|-2.32|-0.21|
> > > |2|-6.12|-5.97|-0.97|
> > > |3|-5.14|-5.01|-0.89|
> > > |4|-3.15|-2.87|-0.64|
> > >
> > > Representation crowding is supported by the above. In the baseline, higher-curvature regions generally exhibit lower ED and much more negative margins, indicating reduced representational capacity and stronger class entanglement. AES provides only limited improvement. In contrast, PointCHR consistently improves margin across all bins and yields clear ED gains in higher-curvature regions. These results show that PointCHR alleviates crowding specifically in geometrically complex regions, rather than merely introducing a generic scaling effect. The slight ED reduction in low-curvature bins further supports that crowding is a relative, curvature-dependent phenomenon, and that our method performs adaptive capacity reallocation rather than uniform feature expansion.
> > >
> > > These findings further highlight the necessity of hyperbolic geometry for 3D point cloud representation. The near-zero δ-hyperbolicity indicates an underlying tree-like structure that is poorly matched by Euclidean embeddings. The ED degradation and margin deterioration observed in Euclidean baselines reveal a fundamental representation bottleneck, namely representation crowding in complex regions. PointCHR mitigates this bottleneck by leveraging the exponential representational capacity of hyperbolic space, providing a more suitable inductive bias for the intrinsic complexity of 3D geometric data.
> > >
> > > [1] V.H.J et al., “Point Cloud Classification with ModelNet40: What Is Left?,” in ICML, 2023.
> > >
> > > [2] M. A. Uy et al., “Revisiting Point Cloud Classification: A New Benchmark Dataset and Classification Model on Real-World Data,” in ICCV, 2019.
> > >
> > > [3] M. Nickel and D. Kiela, “Poincaré Embeddings for Learning Hierarchical Representations,” in NeurIPS, 2017.
> > >
> > > [4] F. Sala et al., “Representation Tradeoffs for Hyperbolic Embeddings,” in ICML, 2018.
> > >
> > > [5] M. G. Atigh et al., “Hyperbolic Image Segmentation,” in CVPR, 2022.
> > >
> > > [6] W. Li et al., “Hyperbolic Hierarchical Alignment Reasoning Network for Text-3D Retrieval,” in AAAI, 2026.

---

### Official Review · Reviewer_e3C6 · 2026-03-09

**Soundness:** 3
**Presentation:** 3
**Significance:** 3
**Originality:** 3
**Overall Recommendation:** 4
**Confidence:** 3

**Summary:**

This paper presents PointCHR, a curvature-aware hyperbolic rectification module designed for point cloud analysis. The core idea is to exploit the exponential volume expansion of the Poincaré ball to mitigate representation crowding in sparse regions with high curvature. Specifically, embeddings of such points are radially expanded toward the high-capacity boundary of the hyperbolic space, guided by point-wise curvature estimation. The proposed approach consists of three main components: (i) Hyperbolic Semantic Transformation, which employs Möbius linear layers and nonlinear activations to perform feature transformations in hyperbolic space; (ii) Point-wise Curvature-Adaptive Perception, a learned gating mechanism that integrates curvature information with point features to generate a radial scaling factor; and (iii) Closed-Form Geodesic Dilation, which performs manifold-consistent radial rescaling without requiring tangent-space projections. Experiments conducted on S3DIS, ShapeNetPart, ModelNet40, and ScanObjectNN demonstrate strong performance, and ablation studies as well as evaluations across multiple backbones confirm the effectiveness and generalization ability of the proposed module..

**Compliance With Llm Reviewing Policy:**

Affirmed.

**Final Justification:**

The authors' rebuttal successfully addressed concerns regarding hyperparameter sensitivity and mathematical consistency, providing clear quantitative evidence of the model's robustness to point density variations. So I maintain my recommendation of Weak Accept.

**Key Questions For Authors:**

1.The hyperbolic distance and mapping equations presented in the paper (e.g., the distance definition and the exponential map) appear to differ from the standard formulation of the Poincaré ball metric, which typically includes an additional scaling factor. Is this discrepancy due to an intentional reparameterization? The authors are encouraged to clarify the equations and confirm that the implementation is consistent with the expressions reported in the paper. If any inconsistency exists, the equations should be corrected and the potential impact on the reported results should be discussed.
2.The PCP module relies on curvature estimates derived from PCA-based k-NN neighborhoods. How robust is this estimation to noisy point normals or variations in point density? Have the authors considered alternative curvature estimation methods or learning-based curvature proxies? It would also be interesting to evaluate how the method behaves when curvature is estimated at multiple neighborhood scales.

**Limitations:**

See Weakness

**Strengths And Weaknesses:**

Strength：
1.The paper introduces a conceptually clean and elegant mechanism that integrates intrinsic curvature cues with hyperbolic embeddings to mitigate representation crowding in point clouds. In particular, the proposed Closed-Form Geodesic Dilation (CGD) is a well-designed operation that adjusts the geodesic radius while keeping points on the manifold, offering greater stability than repeatedly applying logarithmic and exponential mappings. The formulation also unifies geometric-aware gating (PCP) with hyperbolic feature transformations (HST), establishing a principled connection between geometric priors and non-Euclidean representation learning.
2.The method is evaluated on several widely used benchmarks, including S3DIS, ShapeNetPart, ModelNet40, and ScanObjectNN, where it consistently demonstrates performance improvements. The paper provides component-level ablations (e.g., PCP and HST) and plug-and-play experiments across multiple backbone architectures, highlighting the general applicability of the proposed module. Qualitative visualizations are also presented, emphasizing boundary regions where the proposed inductive bias is expected to have the greatest impact.
3.The motivation of the work—namely the long-tailed curvature distribution and representation crowding in Euclidean embeddings—is clearly explained with intuitive visualizations. The overall pipeline and algorithmic procedure are easy to follow, and the three proposed components are well separated, making the method straightforward to understand.
4.The work adopts a geometry-driven perspective to address a recurring failure mode in point cloud segmentation, namely errors occurring in sparse, high-curvature boundary regions. By demonstrating the advantages of hyperbolic representations in 3D point cloud tasks—an area that remains relatively less explored compared to graphs and images—the paper contributes to the growing body of research in hyperbolic deep learning.




Weakness：
1.The paper provides limited ablation studies for several important hyperparameters, including the curvature parameter and other coefficients involved in the proposed modules. In particular, the PCP module relies on a fixed curvature estimate derived from PCA k-NN, yet the sensitivity of the approach to the choice of k, noise levels, or variations in point density is not investigated. Without sensitivity analysis or robustness experiments, it is difficult to evaluate how stable the method is under different data conditions.
2.Although the method introduces mechanisms related to boundary behavior, the paper does not analyze the distribution of the learned scaling factors or report the proportion of points that are pushed close to the boundary of the hyperbolic space. Such analysis would help clarify how the mechanism operates in practice and whether the intended geometric effect is consistently achieved.

---

> ### Author Rebuttal · Authors · 2026-03-29
>
> *R3Q1*:Sensitivity analysis or robustness experiments
>
> **R3A1**:Thank the reviewer for the suggestion. We conduct analyses on key hyperparameters, curvature estimation strategies and noise interference.
>
> 1. Multiple $K$NN scale and $c$ curvature parameter: PointCHR remains stable across different neighborhood sizes, indicating low sensitivity to $K$, also remains stable under different manifold curvatures.
> |$K$|S3DIS mIoU(%)|$c$|S3DIS mIoU(%)|
> |-|-|-|-|
> |16|85.98|0.5|85.87|
> |24|86.01|1|86.01|
> |32|85.97|2|85.93|
>
> 2. Hyperbolic coefficients: A broad range of parameter combinations achieves strong performance, showing robustness to scaling intensity.
> |Alpha\Gamma|1|2|3|
> |-|-:|-:|-:|
> |1|85.88|86.01|85.06|
> |2|85.58|85.66|86.04|
> |3|85.42|85.56|85.00|
>
> 3. Curvature estimation: we conducted a comparative study against the Moving Least Squares (MLS) method. PCA achieves  better performance with significantly lower cost. High order fitting does not yield additional benefit, while introducing large overhead. The PCA based design is efficient and aligns with the lightweight principle.
> |Dataset|Method|mIoU/OA|Time|Epochs|
> |-|-|-:|-:|-:|
> |S3DIS|MLS|85.97|16H|150|
> ||PCA|86.01|10H|150|
> |ScanObjectNN|MLS|92.64|54H|400|
> ||PCA|92.71|34H|400|
>
> 4. Robustness:
> Noise perturbation analysis on **R2A4** indicates strong tolerance to curvature estimation errors and supports stability in noisy settings.  We will include the above in the appendix.
> ---
>
> *R3Q2*:Scaling factor and boundary distribution.
>
> **R3A2**:Thank the reviewer for this valuable suggestion. We analyze the distribution of the learned scaling factor \( s \) across different curvature regimes and spatial regions. The distribution of the learned scaling factor \( s \) across curvature regimes and spatial regions as follows:
>
> | Curvature Bin | Point Ratio (%) | Avg. \(s\) (Non-boundary) | \(s > 1.13\) Non-boundary Ratio  (%) | Avg. \(s\) (Boundary) | \(s > 1.13\) Boundary Ratio (%) |
> |-|-|-|-|-|-|
> | [0.00, 0.05) | 83.64 | 1.13 | 19.81 | 1.22 | 24.18 |
> | [0.05, 0.10) | 10.34 | 1.43 | 40.64 | 1.84 | 53.12 |
> | [0.10, 0.15) | 3.89 | 1.61 | 49.47 | 2.13 | 61.99 |
> | [0.15, 0.20] | 2.13 | 1.76 | 55.50 | 2.13 | 63.00 |
>
> To quantify the proportion of points undergoing relatively strong expansion, we adopt a data-driven reference threshold. Specifically, we set \( s = 1.13 \) as the mean scaling factor of low-curvature non-boundary points, and report the ratio of points with \( s > 1.13 \) as an indicator of enhanced scaling behavior.
>
> The results are summarized in Table 1. We observe a clear positive correlation between local curvature and the magnitude of \( s \). As curvature increases, both the average scaling factor and the proportion of high-scaling points consistently rise. This indicates that PointCHR adaptively assigns stronger scaling to geometrically more complex regions. In addition, boundary points in the scene generally exhibit larger \( s \) values and higher \( s > 1.13 \) ratios than non-boundary points under the same curvature range, suggesting that the proposed mechanism further emphasizes structurally critical regions.
>
> Overall, these results show that PointCHR learns a non-uniform and structure-aware scaling strategy. We will include this in the appendix.
>
> ---
>
> *R3Q3*: The Equations seem inconsistent.
>
> **R3A3**: We thank the reviewer for this careful observation. We clarify that the formulation presented in the manuscript is consistent with our code implementation. The apparent discrepancy comes from a different normalization convention for the hyperbolic distance, where the constant factor 2 is a global scaling term and does not affect the proposed operator. Specifically, our CGD is defined through relative radial scaling:
> $$
> d_c(\mathbf{0}, \mathbf{z}^{(2)}) = s_i \, d_c(\mathbf{0}, \mathbf{z}^{(1)}).
> $$
>
> Under the standard Poincaré-ball formulation, the origin distance is written as
> $$
> d_c(\mathbf{0}, \mathbf{z}) = \frac{2}{\sqrt{c}} \operatorname{arctanh}(\sqrt{c}\|\mathbf{z}\|).
> $$
>
> In our derivation and implementation, the same scaling is applied consistently to both sides. Therefore, the global prefactor 2/√c cancels out. The key point is not that a hyperparameter is ignored, but that the constant factor is a global normalization term. Since CGD is defined through relative radial scaling, this prefactor cancels out in both the derivation and the implementation, and therefore does not affect the learned representation, optimization dynamics, or empirical results. This leads to the same closed-form update used in our method:
> $$
> \mathbf{z}^{(2)}=
> \frac{1}{\sqrt{c}}
> \tanh\left(
> s_i \operatorname{arctanh}(\sqrt{c}\|\mathbf{z}^{(1)}\|)
> \right)
> \frac{\mathbf{z}^{(1)}}{\|\mathbf{z}^{(1)}\|}.
> $$
> Therefore, the constant factor 2 is not a learnable or task-dependent term, but merely a global normalization constant. It does not change the reported empirical results. To avoid ambiguity, we will revise the manuscript to explicitly state this normalization convention.

---

> > ### Author Rebuttal · Reviewer_e3C6 · 2026-04-03
> >
> > Thank you for your detailed and helpful response. The additional analysis regarding hyperparameters, spectral complexity, and the boundary behavior under scaling has addressed most of my concerns. However, regarding the robustness to noise or variations in point density, the current response relies primarily on qualitative discussion and lacks direct experimental verification. Including empirical results in this area would make the claims significantly more persuasive. Overall, the rebuttal is clear and addresses the majority of my questions; therefore, I maintain my previous positive assessment.

---

> > > ### Author Response · Authors · 2026-04-05
> > >
> > > *R3Q1*:The robustness in point density
> > >
> > > **R3A1**: We sincerely thank the reviewer for the positive assessment of our work and for the constructive suggestion. To provide direct empirical verification of robustness to point density variation, we further evaluate test-time robustness under random masking on S3DIS without retraining. Specifically, the model is trained on clean data and directly tested with different masking ratios to simulate point density reduction.
> > >
> > > | Mask ratio | mIoU (%) | Δ |
> > > |-|-|-|
> > > | Clean | 86.01 | - |
> > > | 5% | 86.43 | +0.32 |
> > > | 10% | 86.00 | -0.01 |
> > > | 20% | 86.03 | +0.02 |
> > > | 30% | 85.96 | -0.05 |
> > > | 40% | 84.97 | -1.04 |
> > > | 50% | 83.51 | -2.50 |
> > >
> > > The results show that the model remains highly stable under mild to moderate density variation. Specifically, the performance is nearly unchanged up to a masking ratio of 30%, and noticeable degradation appears only when the masking becomes severe (40%–50%). This indicates that the proposed method preserves discriminative geometric structure well under reduced point density.
> > >
> > > These results provide direct quantitative evidence that our method is robust to point density variation at test time, which makes our previous discussion more convincing.

---

### Official Review · Reviewer_v2GW · 2026-03-11

**Soundness:** 3
**Presentation:** 2
**Significance:** 3
**Originality:** 3
**Overall Recommendation:** 4
**Confidence:** 4

**Summary:**

The manuscript proposes **PointCHR**, a curvature-aware hyperbolic module to address representation crowding in 3D point cloud analysis. It maps Euclidean features into the Poincaré ball and adaptively expands embeddings in high-curvature regions. The design combines Hyperbolic Semantic Transformation, curvature-adaptive perception, and geodesic dilation. Experiments show state-of-the-art results on several 3D benchmarks with strong parameter efficiency.

**Compliance With Llm Reviewing Policy:**

Affirmed.

**Final Justification:**

The rebuttal addresses most of my concerns with stronger baselines, added profiling, and improved robustness analysis, reinforcing my confidence in the work. While evaluation on real outdoor LiDAR data and training efficiency remain minor concerns, they do not outweigh the paper’s strengths. I therefore keep my original positive score unchanged.

**Key Questions For Authors:**

While the method performs well on dense indoor datasets (S3DIS) or CAD models, can it remain robust on extremely sparse or noisy large-scale outdoor point clouds (e.g., LiDAR scans where PCA normals/curvatures are unstable)?

**Limitations:**

yes

**Strengths And Weaknesses:**

### Strengths

- Good Motivation and Novelty: Identifying representation crowding in high-curvature point cloud regions and linking it to the exponential volume growth properties of hyperbolic spaces is a highly intuitive and creative idea.

- Elegant Mathematical Engineering: The introduction of CGD successfully addresses a major pain point in hyperbolic deep learning—numerical instability and arithmetic overflow near the boundaries.

- Strong Performance: The proposed method achieves strong results (e.g., 86.0% mIoU on S3DIS Area 5).

- Plug-and-play Capability: As demonstrated in Table 7, the module generalizes well and improves multiple baseline architectures without requiring heavy architectural modifications.

### Weaknesses

- Lack of Comparison with a Simple Resampling Baseline: The paper motivates its approach by the sparsity of high-curvature points but does not compare against a simple data-level solution such as curvature-guided resampling (e.g., oversampling high-curvature regions instead of FPS). Without this baseline, it is unclear whether the complex hyperbolic pipeline is necessary.

- The performance improvements are somewhat marginal, especially on the classification task.

- Though FLOPs and latency are reported, complex operations such as arctanh, tanh, and exponential maps often introduce unoptimized kernel execution times on standard GPUs. A deeper analysis of memory footprint and actual training overhead is missing.

---

> ### Author Rebuttal · Authors · 2026-03-29
>
> *R2Q1*:Comparison with resampling baseline
>
> **R2A1**:Thank the reviewer for the suggestion. Curvature guided resampling (CGR) is a data level baseline for handling sparsity in high curvature regions. We implement this baseline on S3DIS by using the learned weights in Eq.(12) to prioritize high curvature points.
> |Method|S3DIS mIoU(%)|
> |-|-|
> |Baseline|83.30|
> |CGR|79.54|
> |PointCHR|86.01|
>
> The results show that curvature guided resampling underperforms both the baseline and PointCHR. This indicates that representation crowding cannot be resolved at the data level. Resampling modifies the original point distribution and disrupts spatial continuity and local topology, which degrades the ability of the backbone to model consistent geometric dependencies. Increasing point density in high curvature regions does not improve the intrinsic capacity of the feature space and therefore cannot resolve feature compression.
> PointCHR instead operates in the embedding space and leverages the exponential radial capacity of hyperbolic geometry to perform curvature aware nonlinear feature expansion. This mechanism allocates more representational capacity to geometrically complex regions without modifying spatial structure or discarding contextual information. The results suggest that the limitation lies in feature space capacity rather than point density. A manifold based transformation is therefore more suitable than data level resampling.
>
> ---
>
> *R2Q2*:Limited gains on classification
>
> **R2A2**:Thank the reviewer for the observation. The moderate gains on some classification benchmarks are consistent with the nature of these tasks. The main advantage of PointCHR appears when local geometric complexity is important.
> Datasets such as ModelNet40 are dominated by global shape recognition. Coarse structural cues are sufficient and performance is already saturated. In contrast, PointCHR is designed to alleviate representation crowding in high curvature regions. Therefore, its advantages are more pronounced in tasks that require fine-grained local geometric discrimination.
> This trend is reflected in the experiments. On S3DIS, where accurate local geometry modeling is essential, PointCHR achieves more improvements. This indicates that curvature aware hyperbolic rectification is effective for precise local feature separation. On ModelNet40, PointCHR improves mAcc to 92.0%, suggesting enhanced discrimination for difficult categories. On ScanObjectNN, which contains noise, occlusion, and irregular sampling, PointCHR achieves 92.7% OA and 91.7% mAcc, surpassing recent methods. Although gains on clean global classification tasks are moderate, the consistent improvements across real scan classification and scene understanding demonstrate that PointCHR is effective when local geometric complexity is the dominant factor.
>
> ---
>
> *R2Q3*:Memory and training overhead
>
> **R2A3**:Thank the reviewer for the suggestion. We profile GPU memory usage and training time on S3DIS Area 5 using a single RTX 4090 with 24 GB memory and batch size 8.
> |Method|Memory|Time|S3DIS mIoU(%)|
> |-|-|-|-|
> |Baseline|20.1G|7H|83.30|
> |PointCHR|22.9G|10H|86.01|
>
> PointCHR introduces moderate overhead while providing a clear accuracy gain. Additional memory consumption is mainly due to point wise curvature estimation and intermediate variables in CGD. Increased training time arises from k nearest neighbor based curvature computation and hyperbolic transformations.
> The model remains feasible within a 24 GB GPU, indicating practical applicability. The overhead is reasonable given the 2.71% mIoU improvement. Comparable gains often require larger backbones or more complex architectural changes. We will include this in the appendix.
>
> ---
>
> *R2Q4*: Robustness to noisy sparse dataset.
>
> **R2A4**:Thank the reviewer for the question. Large scale outdoor data often involve noise, and occlusion, which lead to unstable curvature estimation. We simulate these effects through controlled perturbation experiments on S3DIS.
> Gaussian noise is injected into curvature values to simulate sensor noise and nonuniform sampling.
> |σ|S3DIS mIoU(%)|Δ|
> |-|-|-|
> |Clean|86.01|-|
> |0.01|86.37|+0.36|
> |0.05|85.58|-0.43|
> |0.10|85.49|-0.52|
> |0.20|84.78|-1.23|
>
> We also simulate missing curvature by randomly setting curvature values to zero.
> |Ratio|S3DIS mIoU(%)|Δ|
> |-|-|-|
> |Clean|86.01|-|
> |10%|85.84|-0.17|
> |20%|85.61|-0.40|
> |30%|85.10|-0.91|
>
> The results show strong robustness. Performance remains above 85% mIoU even under perturbation and still exceeds the baseline. The closed form CGD mapping provides stable manifold transformation and avoids instability under noisy inputs. Curvature is used as a modulation signal rather than a strict constraint, allowing the backbone to absorb local estimation errors through contextual aggregation.
> These results indicate that PointCHR is intrinsically robust to curvature noise and sparsity. Therefore, we believe the method can generalize well where geometric measurements are imperfect.

---

> > ### Author Rebuttal · Reviewer_v2GW · 2026-04-02
> >
> > Thank you for the detailed rebuttal. The new resampling baseline, the added memory/training profiling, and the robustness analyses address most of my original concerns and strengthen the paper. I still would have liked to see evidence on truly outdoor large-scale point clouds rather than perturbation-based simulations on S3DIS, since that would better support the claim of robustness in realistic LiDAR settings. I also think the reported training overhead is not negligible in practice, as the training time increases from 7 hours to 10 hours even if the memory remains manageable. Overall, however, the rebuttal is solid and resolves most of my questions, so I maintain my positive scores.

---

> > > ### Author Response · Authors · 2026-04-08
> > >
> > > Thank you for your constructive feedback and for maintaining your positive support of our work. We are encouraged that our new baselines and robustness analyses successfully addressed most of your concerns.
> > >
> > > Regarding your further suggestions, we would like to provide the following clarifications:
> > >
> > > Outdoor Large-scale Experiments: We agree that evaluating on real-world LiDAR datasets (such as Waymo) would further strengthen our claims of robustness. Due to the significant computational resources required for processing such large-scale data and our current hardware constraints, we were unable to complete these experiments within the short rebuttal window. However, we are actively conducting these evaluations and commit to including the results in the final version of the paper.
> > >
> > > Training Overhead: Your point regarding the training time increase (from 7h to 10h) is well-taken. We will add a dedicated discussion in the final manuscript to acknowledge this trade-off.
> > >
> > > Thank you again for your insightful comments, which are helping us make the paper more robust and practical.

---

### Official Review · Reviewer_1Vun · 2026-03-12

**Soundness:** 3
**Presentation:** 3
**Significance:** 2
**Originality:** 2
**Overall Recommendation:** 4
**Confidence:** 4

**Summary:**

The paper proposes PointCHR, a curvature-aware hyperbolic rectification module for point cloud analysis. The approach maps Euclidean features to the Poincare ball, computes a pointwise curvature-aware scaling that depends on estimated curvature and learned gating, and then performs a closed-form geodesic dilation to push high-curvature points radially toward the boundary before mapping back to Euclidean space. Integrated with CamPoint and other backbones, the method improves semantic segmentation on S3DIS, part segmentation on ShapeNetPart, and classification on ModelNet40 and ScanObjectNN.

**Compliance With Llm Reviewing Policy:**

Affirmed.

**Key Questions For Authors:**

See weakness.

**Limitations:**

Yes.

**Strengths And Weaknesses:**

Strength:
1. Curvature-conditioned radial rectification is simple yet potentially influential. The s_i definition (Equation 13) couples a learned gate with curvature, which is an interpretable mechanism that could be re-used in other hyperbolic or non-Euclidean setups.
2. Breadth of evaluation across tasks and datasets. Results cover S3DIS, ShapeNetPart, ModelNet40, and ScanObjectNN, suggesting applicability beyond a single benchmark.
3. Presentation is generally clear, with a helpful block diagram and convincing qualitative segmentation in Figure 3 where boundary delineation visibly improves vs. CamPoint.

Weakness:
1. Novelty and positioning relative to prior hyperbolic and curvature-aware point cloud work are underdeveloped. While the combination is interesting, there is prior art (see references) on hyperbolic embeddings for point clouds and curvature-aware modeling. The paper cites hyperbolic vision and graphs, but omits directly relevant point cloud works that already explored hyperbolic features or curvature-driven mechanisms.
2. Stability and optimization claims are asserted but not empirically supported. The paper argues CGD avoids vanishing or overflow near the boundary. However, there is no study of gradient norms, training curves, or failure modes as s_i increases
3. The improvements on ModelNet40 and ShapeNetPart are small (Tables 3, 4). The significance or variability is not discussed.
4. Section 5.1 misses practical training details needed for reproducibility: batch size, epochs, learning rate values, data augmentations per dataset, and how curvature is normalized to [0,1].

Reference
1. Rethinking the Compositionality of Point Clouds through Regularization in the Hyperbolic Space.
2. Hyperbolic Prototype Rectification for Few-Shot 3D Point Cloud Classification
3. CanonNet: Canonical Ordering and Curvature Learning for Point Cloud Analysis

---

> ### Author Rebuttal · Authors · 2026-03-31
>
> *R1Q1*:The positioning is unclearly articulated.
>
> **R1A1**:Thank the reviewer for this important comment. We will revise related-work accordingly.
>
> Existing hyperbolic methods mainly use hyperbolic geometry to model hierarchical or prototype-level relationships. For example, *HyCoRe* regularizes point-cloud representations in hyperbolic space to better capture part-whole compositional hierarchy, while *HPR* uses hyperbolic embeddings and prototype rectification for few-shot classification. In both cases, hyperbolic space is primarily introduced to encode semantic hierarchy, rather than to allocate point-wise representational capacity according to local geometric complexity.
>
> Existing curvature-aware methods mainly exploit curvature for geometric feature learning or supervision. For instance, *CanonNet* combines canonical ordering/orientation with curvature-based synthetic data generation to learn fine-grained geometric properties, focusing on curvature estimation and geometric descriptor learning rather than dynamic feature-space capacity control.
>
> In contrast, the innovation of *PointCHR* lies in the alignment of curvature and capacity: local geometric complexity is explicitly mapped to hyperbolic radial capacity, thereby unfolding high-curvature points into regions of higher capacity on the Poincaré ball. This mechanism is achieved through PCP and CGD, representing a fundamental departure from previous hyperbolic hierarchical modeling approaches and prior curvature-aware geometric learning methods.
>
> ---
>
> *R1Q2*:The stability of CGD need discussed.
>
> **R1A2**:Thank the reviewer for the insightful concern. We note that other reviewers have also acknowledged the robustness of our formulation, and we take this opportunity to clarify the stability of CGD from both theoretical and empirical perspectives.
>
> Our CGD adopts a closed-form formulation with explicit numerical safeguards. The CGD gradient behavior is governed by
> $$
> \frac{\partial}{\partial r} \operatorname{arctanh}(\sqrt{c}r) = \frac{\sqrt{c}}{1 - c r^2}.
> $$
> Thus, instability arises only when $r \to 1/\sqrt{c}$. By explicitly constraining $\tilde r_i < (1-\epsilon)/\sqrt{c}$, we ensure that $1 - c r^2$ remains strictly positive, leading to bounded gradients. Importantly, the use of the closed-form mapping avoids the log-exp transformation chain, reducing numerical amplification and improving optimization stability.
>
> We further clarify that stability is fundamentally determined by the radial norm $r=\|x\|$, rather than the scaling factor $s_i$. While $s_i$ linearly scales the geodesic distance, it does not introduce singularities; it only amplifies gradients when $r$ is already close to the boundary.
>
> We independently validated this design five times on S3DIS. Without stability control, training exhibits numerical issues. The safe formulation not only eliminates numerical instability, but also achieves higher performance with significantly lower variance, indicating both improved accuracy and training stability.
> |Method|Inf Count|mIoU(mean ± std)|
> |-|-|-|
> |Safe|0|86.00 ± 0.22|
> |Unsafe|24|84.73 ± 0.79|
>
> Our formulation does not eliminate the intrinsic singularity of hyperbolic geometry, but makes it explicitly controllable via norm constraints and closed-form mapping, ensuring stable and reliable optimization.
>
> ---
>
> *R1Q3*:The significance need discussed.
>
> **R1A3**:Thank the reviewer for this observation. The reported results are obtained from 5 independent runs, showing high stability (**93.68±0.06% on ModelNet40** and **86.95±0.09% on ShapeNetPart**), confirming that the gains are consistent rather than due to randomness. The relatively limited improvement on these benchmarks is expected, as they are near-saturated and dominated by low-curvature global structures, where Euclidean models already perform well and leave limited room for improvement. In contrast, PointCHR is specifically designed to enhance representation capacity in high-curvature local regions, whose contribution is underrepresented in OA.
>
> ---
>
> *R1Q4*:Lack of parameters.
>
> **R1A4**:Thank the reviewer for the constructive suggestion. Our implementation follows the same settings as CamPoint. We will include the parameters in the appendix. The key training details below:
>
> |Dataset|Batch Size|Epochs|LR|Scheduler|Warmup|
> |-|-|-|-|-|-|
> |S3DIS|8|150|1e-3|Cosine decay|10|
> |ShapeNetPart|32|400|2e-3|Cosine decay|30|
> |ScanObjectNN|32|400|5e-4|Cosine decay|30|
> |ModelNet40|32|400|5e-4|Cosine decay|30|
>
> The augmentations, including random rotation, scaling within [0.8,1.2], and random point dropout (10%), to improve robustness.
> To map raw curvature κ into a stable range, we apply a min–max normalization followed by clipping:
> $$
> \hat{\kappa} = \mathrm{clamp}\left(\frac{\kappa - \kappa_{\min}}{\kappa_{\max} - \kappa_{\min}}, 0, 1\right).
> $$
> In practice, we set $κ_{max}$=0.2 as a threshold to suppress extreme outliers, ensuring stable estimation of the scaling factor $s_{i}$.

---

> > ### Author Rebuttal · Reviewer_1Vun · 2026-04-03
> >
> > Thank you for your thorough and well-structured rebuttal. You have addressed my major concerns especially the stability of CGD is now well explained with theoretical analysis.The significance discussion convincingly explains why near‑saturated benchmarks show limited gains and highlights the real value in high‑curvature local regions.
> > I will maintain my positive scores.

---

> > > ### Author Response · Authors · 2026-04-08
> > >
> > > Thank you very much for your positive feedback and for maintaining your support for our work.Your insightful comments during the review process have been instrumental in improving the quality and clarity of our manuscript.We will ensure that these additional analyses and discussions are fully integrated into the final version of the paper.

---

### Decision · Program_Chairs · 2026-04-30

**Decision:**

Accept (regular)

**Comment:**

This paper presents PointCHR, a curvature-aware hyperbolic rectification module designed to alleviate "representation crowding" in 3D point cloud analysis by mapping Euclidean features to the Poincaré ball. The reviewers point out that the method is conceptually elegant and technically solid, offering a plug-and-play solution that improves performance across semantic segmentation and classification benchmarks. The paper receives ratings of four weak accepts from these reviewers. Considering the reviewers' comments and the authors' rebuttal, the AC remains convinced of the work's contribution. The AC decides to accept the paper and encourages the authors to incorporate the rebuttal’s analyses and discussions into the final manuscript.